# Reward Guided Latent Consistency Distillation

**Jiachen Li**                                                    *jiachen_li@cs.ucsb.edu*
*University of California, Santa Barbara*

**Weixi Feng**                                                    *weixifeng@cs.ucsb.edu*
*University of California, Santa Barbara*

**Wenhu Chen**                                                    *wenhu.chen@uwaterloo.ca*
*University of Waterloo*

**William Yang Wang**                                             *william@cs.ucsb.edu*
*University of California, Santa Barbara*

**Reviewed on OpenReview:** *https://openreview.net/forum?id=z116TO4LDT*

## Abstract

Latent Consistency Distillation (LCD) has emerged as a promising paradigm for efficient text-to-image synthesis. By distilling a latent consistency model (LCM) from a pre-trained teacher latent diffusion model (LDM), LCD facilitates the generation of high-fidelity images within merely 2 to 4 inference steps. However, the LCM's efficient inference is obtained at the cost of the sample quality. In this paper, we propose compensating the quality loss by aligning LCM's output with human preference during training. Specifically, we introduce Reward Guided LCD (RG-LCD), which integrates feedback from a reward model (RM) into the LCD process by augmenting the original LCD loss with the objective of maximizing the reward associated with LCM's single-step generation. As validated through human evaluation, when trained with the feedback of a good RM, the 2-step generations from our RG-LCM are favored by humans over the 50-step DDIM (Song et al., 2020a) samples from the teacher LDM, representing a 25-time inference acceleration without quality loss.

As directly optimizing towards differentiable RMs can suffer from over-optimization, we take the initial step to overcome this difficulty by proposing the use of a latent proxy RM (LRM). This novel component serves as an intermediary, connecting our LCM with the RM. Empirically, we demonstrate that incorporating the LRM into our RG-LCD successfully avoids high-frequency noise in the generated images, contributing to both improved Fréchet Inception Distance (FID) on MS-COCO (Lin et al., 2014) and a higher HPSv2.1 score on HPSv2 (Wu et al., 2023a)'s test set, surpassing those achieved by the baseline LCM.

Project Page: `https://rg-lcd.github.io/`

## 1 Introduction

In the realm of modern generative AI (GenAI) models, computational resources are typically allocated across three key areas: pretraining (Brown et al., 2020; Achiam et al., 2023a; Li et al., 2022b; Radford et al., 2021; Rombach et al., 2022; Saharia et al., 2022; Betker et al., 2023a), alignment (Ziegler et al., 2019; Stiennon et al., 2020; Ouyang et al., 2022; Hu et al., 2024; Clark et al., 2023; Rafailov et al., 2024), and inference (Zhang & Chen, 2022; Feng et al., 2023; Vijayakumar et al., 2016; Shih et al., 2024). Normally, increasing the computational budget across these areas leads to improvements in sample quality. For instance, the most advanced text-to-image (T2I) models, such as DALLE-3 (Betker et al., 2023a), Imagen (Saharia et al., 2022), and Stable Diffusion (Rombach et al., 2022) are built from diffusion models (DMs) (Sohl-Dickstein et al., 2015; Ho et al., 2020; Song & Ermon, 2019). These models are pretrained on massive web-scale

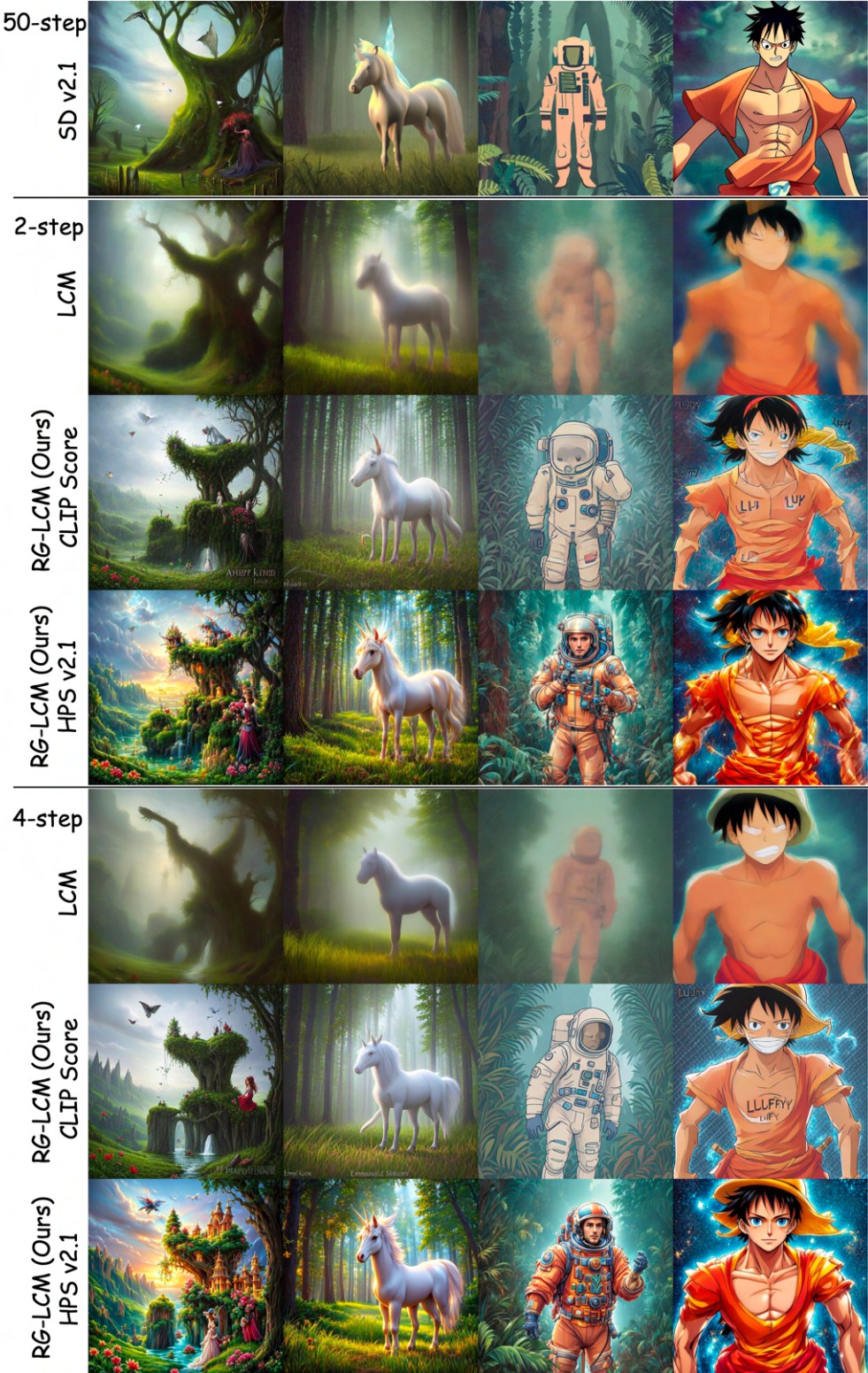

Figure 1: Even with merely 2-4 sampling steps, our RG-LCMs that learned from the CLIP Score and HPSv2.1 can produce high-quality images.

datasets (Schuhmann et al., 2022; Changpinyo et al., 2021), aligned with human preference on curated high-quality images (Dai et al., 2023; Rombach et al., 2022), and benefit from DMs' iterative sampling process.

However, DM's iterative sampling requires performing 10 - 2000 sequential function evaluations (FEs) (Ho et al., 2020; Song et al., 2020a), thus impeding rapid inference. While there have been many works proposed to address this issue (Lu et al., 2022a;b; Zhang et al., 2023; Sauer et al., 2023; Geng et al., 2024; Nguyen & Tran, 2023; Song et al., 2023), consistency model (CM) (Song et al., 2023) emerges as a new family of GenAI model to facilitate fast sampling. Specifically, a CM is trained to perform single-step generation while supporting multi-step sampling to trade compute for sample quality. We can distill a CM from a pretrained DM, a process known as consistency distillation (CD). For instance, Luo et al. (2023a) distill a Latent CM (LCM) from a pretrained Stable Diffusion (Rombach et al., 2022), achieving high-fidelity image generation in just 2 to 4 FE steps. However, the sample quality of LCM is inherently constrained by the pretrained LDM's capabilities (Song & Dhariwal, 2023). Additionally, the reduced inference computational resources stemming from the limited number of FE steps compromise LCM's sample quality.

In this paper, we aim to offset LCM's sample quality by dedicating additional computational resources to the training process. Recent advancements in large language models (Achiam et al., 2023b; Team et al., 2023) have shown that aligning a GenAI model with a reward model (RM) that mirrors human preferences can substantially improve sample quality by reducing undesirable outputs (Ouyang et al., 2022; Rafailov et al., 2024). Thus, we are motivated to align the learned LCM with human preferences by optimizing towards off-the-shelf text-image RMs. Instead of designing a separate alignment phase, we leverage the single-step generation that naturally arises from computing the LCD loss and implement a training objective to maximize its associated rewards given by a differentiable RM through gradient descent. Notably, our approach obviates the need for backpropagating gradients through the complicated denoising procedures, which is typically required by previous methods when optimizing a DM (Clark et al., 2023; Xu et al., 2024; Prabhudesai et al., 2023; Yang et al., 2024). We dub our method *Reward Guided Latent Consistency Distillation* (RG-LCD). Human evaluation shows that our RG-LCM significantly outperforms the LCM derived from standard LCD. Remarkably, our 2-step generation is favored by humans over the 50-step generation from the teacher LDM, representing a 25-fold inference acceleration without compromising image quality.

While our RG-LCD is conceptually simple and already achieves impressive results, it can suffer from reward overestimation (Kim et al., 2023b; Zhang et al., 2024) due to direct optimization with the gradient from the RM. As shown in the top row of Fig. 3, performing RG-LCD with ImageReward (Xu et al., 2024) causes high-frequency noise in the generated images. In this paper, we take an initial step to tackle this challenge. We propose learning a latent proxy RM to serve as the intermediary that connects our LCM with the RM. Instead of directly optimizing towards the RM, we optimize the LCM towards the LRM while finetuning the LRM to match the preference of the expert RM in each RG-LCD iteration. This novel strategy allows us to optimize the expert RM indirectly, even allowing for learning from non-differentiable RMs. We empirically verify that incorporating the LRM into our RG-LCD successfully eliminates the high-frequency noise in the generated image, contributing to improved FID on MS-COCO (Lin et al., 2014) and a higher HPSv2.1 score on HPSv2's test set (Wu et al., 2023a), outperforming the baseline LCM.

In summary, our contributions are threefold:

- Introduction of RG-LCD framework, which incorporates feedback from an RM that mirrors human preference into the LCD process.

- Introduction of the LRM, which enables indirect optimization towards the RM, mitigating the issue of reward over-optimization.

- A 25 times inference acceleration over teacher LDM (Stable Diffusion v2.1) without compromising sample quality.

## 2 Related Work

**Accelerating DM inference**. Centering around DM's SDE formulation (Song et al., 2020b), various methods have been proposed to accelerate the sampling process of a DM. For example, faster numerical

ODE solvers (Song et al., 2020a; Lu et al., 2022a;b; Zheng et al., 2022; Dockhorn et al., 2022; Jolicoeur-Martineau et al., 2021) and distillation techniques (Luhman & Luhman, 2021; Salimans & Ho, 2022; Meng et al., 2023; Zheng et al., 2023). Recent advances explore enhancing the single-step generation quality by incorporating an adversarial loss (Sauer et al., 2023) or by distillation (Nguyen & Tran, 2023). Consistency Model (Song et al., 2023) is also trained for single-step generation. We leverage this property and directly maximize the reward of this single-step generation given by a differentiable RM, avoiding the complexities of backpropagating gradients through the iterative sampling process of a DM.

**Consistency Model** has emerged as a new family of GenAI model (Song et al., 2023) that facilitates fast inference. While it is trained to perform single-step generation by mapping arbitrary points in the PF-ODE trajectory to the origin, CM also supports multi-step sampling, allowing for trading compute for better sample quality. On the one hand, a CM can be trained as a standalone GenAI model (consistency training). Recently, Song & Dhariwal (2023) proposed improved techniques to support better consistency training. On the other hand, a CM can also be distilled from a pretrained DM (Kim et al., 2023a). For instance, Luo et al. (2023a) learn an LCM by distilling from a pretrained Stable Diffusion (Rombach et al., 2022). We defer more technical details to Sec. 3.

**Vision-and-language reward models.** Motivated by the significant success of reinforcement from human feedback (RLHF) in training the LLMs, there have been many works delving into training an RM to mirror human preferences on a pair of text and image, including HPSv1 (Wu et al., 2023b), HPSv2 (Wu et al., 2023a), ImageReward (Xu et al., 2024), and PickScore (Kirstain et al., 2024). These RMs are normally derived by finetuning a vision-and-language foundation model, e.g., CLIP (Radford et al., 2021) and BLIP (Li et al., 2022a), on human preference data. Since these RMs are differentiable, our RG-LCD augments the standard LCD with the objective of maximizing the differentiable reward associated with its single-step generation during training.

**Aligning DMs to Human preference** has been extensively studied recently, including RL based methods (Fan et al., 2024; Prabhudesai et al., 2023; Zhang et al., 2024) and reward finetuning methods (Clark et al., 2023; Xu et al., 2024). Recently, Diffusion-DPO (Wallace et al., 2023a) is proposed by extending DPO (Rafailov et al., 2024) to train DMs on preference data. Moreover, other works focus on modifying the training data distribution (Wu et al., 2023b; Lee et al., 2023; Dong et al., 2023; Sun et al., 2023; Dai et al., 2023) to finetune DMs on visually appealing and textually cohered data. Additionally, alternative techniques (Betker et al., 2023b; Segalis et al., 2023) re-caption pre-collected web images to enhance text accuracy. On the other hand, DOODL (Wallace et al., 2023b) is proposed to optimize the RM during inference time. However, its improvement is made at the cost of inference speed. While we also propose to directly finetune our model with the gradient given by an RM, finetuning an LCM during LCD is much simpler than finetuning a DM, as we only tackle the single-step generation, circumventing the need to pass gradients through the complicated iterative sampling process of a DM.

## 3 Background

### 3.1 Diffusion Model

Diffusion models (DMs) (Sohl-Dickstein et al., 2015; Song & Ermon, 2019; Ho et al., 2020; Nichol & Dhariwal, 2021) progressively inject Gaussian noise into data in the forward process and sequentially denoise the data to create samples in the reverse denoising process. The forward process perturbs the original data distribution $p_{data}(\mathbf{x}) \equiv p_0(\mathbf{x}_0)$ to the marginal distributional $p_t(\mathbf{x}_t)$. From a continuous-time perspective, we can represent the forward process with a stochastic differential equation (SDE) (Song et al., 2020b; Karras et al., 2022)

$$d\mathbf{x}_t = \boldsymbol{\mu}(t)\mathbf{x}_t dt + \sigma(t)d\mathbf{w}_t, \quad \mathbf{x}_0 \sim p_{\text{data}}(\mathbf{x}_0), \tag{1}$$

where $\boldsymbol{\mu}(\cdot)$ and $\sigma(\cdot)$ are the drift and diffusion coefficients respectively, and $\mathbf{w}_t$ denotes the standard Wiener process. The reverse time SDE above corresponds to an ordinary differential equation (ODE) (Song et al., 2020b), named Probability Flow (PF-ODE), which is given by

$$d\mathbf{x}_t = \left[\boldsymbol{\mu}(t)\mathbf{x}_t - \frac{1}{2}\sigma(t)^2 \nabla \log p_t(\mathbf{x}_t)\right] dt, \quad \mathbf{x}_T \sim p_T(\mathbf{x}_T). \tag{2}$$

PF-ODE's solution trajectories sampled at $t$ are distributed the same as $p_t(\mathbf{x}_t)$. Empirically, we learn a denoising model $\boldsymbol{\epsilon}_\theta(\mathbf{x}_t, t)$ to fit $-\nabla \log p_t(\mathbf{x}_t)$ (score function) via score matching (Hyvärinen & Dayan, 2005; Song & Ermon, 2019; Ho et al., 2020). During sampling, we start from the sample $\mathbf{x}_T \sim \mathcal{N}(\mathbf{0}, \tilde{\sigma}^2\mathbf{I})$ and follow the empirical PF-ODE below

$$d\mathbf{x}_t = \left[ \boldsymbol{\mu}(t)\,\mathbf{x}_t + \frac{1}{2}\sigma(t)^2 \boldsymbol{\epsilon}_\theta(\mathbf{x}_t, t) \right] dt, \quad \mathbf{x}_T \sim \mathcal{N}(\mathbf{0}, \tilde{\sigma}^2\mathbf{I}). \tag{3}$$

In this paper, we focus on conditional LDM that operates on the image latent space $\mathcal{Z}$ and includes a text prompt $\mathbf{c}$ passed to the denoising model $\boldsymbol{\epsilon}_\theta(\mathbf{z}_t, \mathbf{c}, t)$, where $\mathbf{z}_t = \mathcal{E}(\mathbf{x}_t) \in \mathcal{Z}$ is encoded by a VAE (Kingma et al., 2021) encoder $\mathcal{E}$. Moreover, we utilize Classifier-Free Guidance (CFG) (Ho & Salimans, 2022) to improve the quality of conditional sampling by replacing the noise prediction with a linear combination of conditional and unconditional noise prediction for denoising, i.e., $\tilde{\boldsymbol{\epsilon}}_\theta(\mathbf{z}_t, \omega, \mathbf{c}, t) = (1+\omega)\boldsymbol{\epsilon}_\theta(\mathbf{z}_t, \mathbf{c}, t) - \omega\boldsymbol{\epsilon}_\theta(\mathbf{z}, \varnothing, t)$, where $\omega$ is the CFG scale.

## 3.2 Consistency Model

Consistency model (CM) (Song et al., 2023) is proposed to facilitate efficient generation. At its core, CM learns a consistency function $\boldsymbol{f} : (\mathbf{x}_t, t) \mapsto \mathbf{x}_\epsilon$ that can map any point $\mathbf{x}_t$ on the same PF-ODE trajectory to the trajectory's origin, where $\epsilon$ is a fixed small positive number. Learning the consistency function involves enforcing the *self-consistency* property

$$\boldsymbol{f}(\mathbf{x}_t, t) = \boldsymbol{f}(\mathbf{x}'_t, t'), \forall t, t' \in [\epsilon, T], \tag{4}$$

where $\mathbf{x}_t$ and $\mathbf{x}'_t$ belong to the same PF-ODE. The consistency function $\boldsymbol{f}$ is modeled with a CM $\boldsymbol{f}_\theta$. To ensure $\boldsymbol{f}_\theta(\mathbf{x}, \epsilon) = \mathbf{x}$, $\boldsymbol{f}_\theta$ is parameteried as

$$\boldsymbol{f}_\theta(\mathbf{x}, t) = c_{\text{skip}}(t)\mathbf{x} + c_{\text{out}}(t)F_\theta(\mathbf{x}, t), \tag{5}$$

where $c_{\text{skip}}(t)$ and $c_{\text{out}}(t)$ are differentiable functions with $c_{\text{skip}}(\epsilon) = 1$ and $c_{\text{out}}(\epsilon) = 0$, and $F_\theta$ is a neural network. We can learn a CM $\boldsymbol{f}_\theta$ by distilling from a pretrained DM, known as *consistency distillation* (CD) (Song et al., 2023). The CD loss is given by

$$L_{\text{CD}}\left(\theta, \theta^-; \Phi\right) = \mathbb{E}_{\mathbf{x}, t}\left[ d\left( \boldsymbol{f}_\theta\left(\mathbf{x}_{t_{n+1}}, t_{n+1}\right), \boldsymbol{f}_{\theta^-}\left(\hat{\mathbf{x}}_{t_n}^\phi, t_n\right) \right) \right]. \tag{6}$$

where $d(\cdot, \cdot)$ measures the distance between two samples. $\theta^-$ is the parameter of the target CM $\boldsymbol{f}_{\theta^-}$, updated by the exponential moving average (EMA) of $\theta$, i.e., $\theta^- \leftarrow \texttt{stop\_grad}(\mu\theta + (1-\mu)\theta^-)$. $\hat{\mathbf{x}}_{t_n}^\phi$ is an estimation of $\mathbf{x}_{t_n}$ from $\mathbf{x}_{t_{n+1}}$ using the one-step ODE solver $\Phi$:

$$\hat{\mathbf{x}}_{t_n}^\phi \leftarrow \mathbf{x}_{t_{n+1}} + (t_n - t_{n+1})\,\Phi\left(\mathbf{x}_{t_{n+1}}, t_{n+1}; \phi\right). \tag{7}$$

Note that the parameter $\phi$ corresponds to the parameter of the pretrained DM, which is used to construct the ODE solver $\Phi$. We includes the algorithm for sampling from a learned CM in Appendix B.

## 3.3 Latent Consistency Model

Luo et al. (Luo et al., 2023a) extends CM to work on latent space $\mathcal{Z}$ and focuses on conditional generation. Specifically, a Latent CM (LCM) $\boldsymbol{f}_\theta : (\mathbf{z_t}, \omega, \boldsymbol{c}, t) \mapsto \mathbf{z_0}$ is trained to minimized the LCD loss

$$L_{\text{LCD}}\left(\theta, \theta^-; \Psi\right) = \mathbb{E}_{\mathbf{z}, \mathbf{c}, \omega, n}\left[ d\left( \boldsymbol{f}_\theta\left(\mathbf{z}_{t_{n+k}}, \omega, \mathbf{c}, t_{n+k}\right), \boldsymbol{f}_{\theta^-}\left(\hat{\mathbf{z}}_{t_n}^{\Psi, \omega}, \omega, \mathbf{c}, t_n\right) \right) \right], \tag{8}$$

where $\hat{\mathbf{z}}_{t_n}^{\Psi, \omega}$ is an estimate of $z_{t_n}$ obtained by the numerical augmented PF-ODE solver $\Psi$ and $k$ is skipping interval.

$$\hat{\mathbf{z}}_{t_n}^{\Psi, \omega} \leftarrow \mathbf{z}_{t_{n+k}} + (1+\omega)\Psi(\mathbf{z}_{t_{n+k}}, t_{n+k}, t_n, c; \psi) - \omega\Psi(\mathbf{z}_{t_{n+k}}, t_{n+k}, t_n, \varnothing; \psi). \tag{9}$$

In this paper, we use DDIM (Song et al., 2020a) as the ODE solver $\Psi$ to distill from a pretrained Stable Diffusion (Rombach et al., 2022) and refer interested readers to the original LCM paper for formula of the DDIM solver. We use huber loss as our distance function $d(\cdot, \cdot)$.

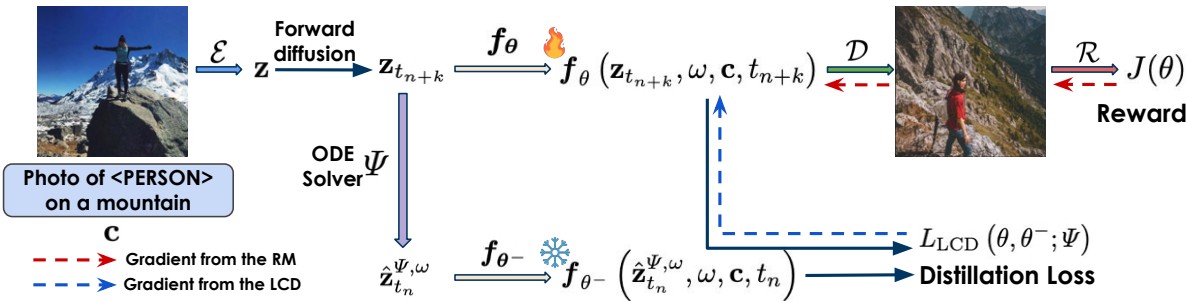

Figure 2: Overview of our RG-LCD. We integrate feedback from a differentiable RM into the standard LCD procedures by training the LCM to maximize the reward associated with its single-step generation..

# 4 Reward Guided Latent Consistency Distillation

In this section, we start by presenting the core components of our RG-LCD framework, which augments the standard LCD loss equation 8 with an objective towards maximizing a differentiable RM, as shown in Fig. 2 (Sec. 4.1). We then motivate the development of a latent proxy RM (LRM) to support indirect RM optimization by illustrating the risk of suffering from reward over-optimization when directly optimizing towards the RM with a gradient-based method. Following this, we then detail the procedure to pretrain and finetune the LRM to match the preference of the RGB-based RM during RG-LCD (Sec. 4.2).

## 4.1 RG-LCD with Differentiable RMs

Recall that each LCD iteration samples a timestep $t_{n+k}$, and construct the noisy latent $\mathbf{z}_{t_{n+k}}$ by perturb the image latent $\mathbf{z} = \mathcal{E}(\mathbf{x})$ with a Gaussian noise, given a sampled CFG scale $\omega$ and text prompt $\mathbf{c}$. As the LCM $\boldsymbol{f}_\theta$ maps the $\mathbf{z}_{t_{n+k}}$ to the PF-ODE origin $\hat{\mathbf{z}}_0 = \boldsymbol{f}_\theta \left( \mathbf{z}_{t_{n+k}}, \omega, \mathbf{c}, t_{n+k} \right)$, we construct the following objective to maximize the reward associated with $\mathcal{D}(\hat{\mathbf{z}}_0)$

$$J(\theta) = \mathbb{E}_{\mathbf{z}, \mathbf{c}, \omega, n} \left[ \mathcal{R} \left( \mathcal{D} \left( \boldsymbol{f}_\theta \left( \mathbf{z}_{t_{n+k}}, \omega, \mathbf{c}, t_{n+k} \right) \right), \mathbf{c} \right) \right], \tag{10}$$

where $\mathcal{R}$ is a differentiable RM that calculates the rewards associated with a pair of text and image. We define the training loss of our RG-LCD by a linear combination of the LCD loss in equation 8 and $J(\theta)$ with a weighting parameter $\beta$

$$L_{\text{RG-LCD}} \left( \theta, \theta^-; \Psi \right) = L_{\text{LCD}} \left( \theta, \theta^-; \Psi \right) - \beta J(\theta) \tag{11}$$

Appendix B includes pseudo-codes for our RG-LCD training.

## 4.2 RG-LCD with a Latent Proxy RM

When training the LCM $\boldsymbol{f}_\theta$ towards $J(\theta)$ with a gradient-based method, we may suffer from the issue of reward over-optimization. As shown in the top row of Fig. 3, performing RG-LCD with ImageReward (Xu et al., 2024) causes high-frequency noise in the generated images. To mitigate this issue, we propose learning a latent proxy RM $\mathcal{R}_\sigma^{\text{L}}$ to serve as an intermediary to connect $\boldsymbol{f}_\theta$ and the expert RGB-based RM $\mathcal{R}^{\text{E}}$, where the $E$ stands for "Expert". Specifically, we train $\boldsymbol{f}_\theta$ to optimize the reward given by $\mathcal{R}_\sigma^{\text{L}}$ while simultaneously finetuning the $\mathcal{R}_\sigma^{\text{L}}$ to matches the preference given by the expert RM $\mathcal{R}^{\text{E}}$ that process RGB images.

Ideally, the LRM $\mathcal{R}_\sigma^{\text{L}}$ should be capable of accessing the text-image pair even at the beginning of RG-LCD. We thus initialize $\mathcal{R}_\sigma^{\text{L}}$ with a pretrained CLIP (Radford et al., 2021) text encoder, complemented by pretraining its latent encoder from scratch. This latent encoder is pretrained following the same methodology used for CLIP visual encoders, ensuring it aligns effectively with the text encoder's representation.

After pretraining, we finetune $\mathcal{R}_\sigma^{\text{L}}$ to match the preference of $\mathcal{R}^{\text{E}}$. Note that we do not need to assume a differentiable $\mathcal{R}^{\text{E}}$ anymore, allowing us to learn from the feedback from a wider range of RGB-based

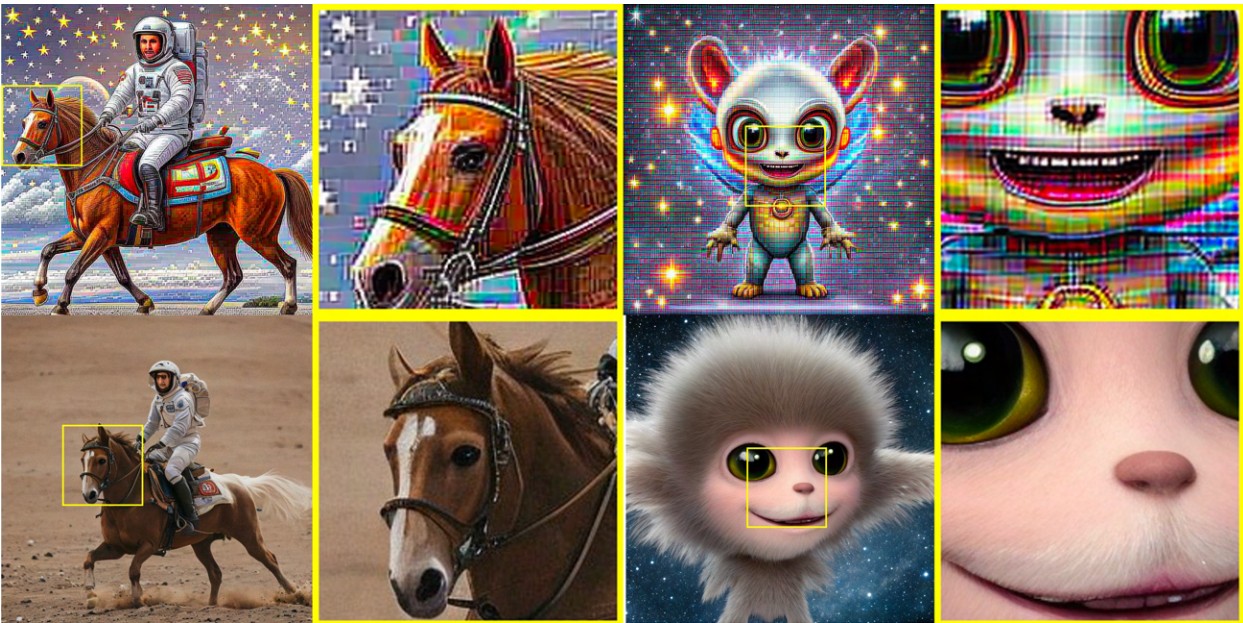

An astronaut riding a horse.

A cute fluffy sentient alien from planet Axor, in the andromeda galaxy, the alien have large innocent eyes and is digitigrade, high detail.

Figure 3: (Top) Optimizing the RG-LCM with the gradient from ImageReward (Xu et al., 2024) results in high-frequency noise in the generated images. (Bottom) Indirectly optimizing the ImageReward through the latent proxy RM eliminates the high-frequency noise, avoiding reward over-optimization.

RM, e.g., LLMScore (Lu et al., 2024), VIEScore (Ku et al., 2023) and DA-score (Singh & Zheng, 2024). Next, we will derive the finetuning loss for our $L_{\mathrm{RM}}(\sigma)$. Given $\mathbf{z}_0 = \mathbf{z}$, $\mathbf{z}_1 = \boldsymbol{f}_\theta\left(\mathbf{z}_{t_{n+k}}, \omega, \mathbf{c}, t_{n+k}\right)$, and $\mathbf{z}_2 = \boldsymbol{f}_{\theta^-}\left(\mathbf{z}_{t_n}, \omega, \mathbf{c}, t_n\right)$ in each RG-LCD iteration, we can group them into three pairs: $(\mathbf{z}_0, \mathbf{z}_1)$, $(\mathbf{z}_0, \mathbf{z}_2)$ and $(\mathbf{z}_1, \mathbf{z}_2)$. We then use $R_\sigma^L$ and $R^E$ to compute the rewards for each latent. For each latent pair $(\mathbf{z}_i, \mathbf{z}_j)$, the probability of $R_\sigma^L$ preferring $\mathbf{z}_i$ over $\mathbf{z}_j$ is modeled as:

$$P_{i,j}^\sigma(i) = \frac{\exp\left(\mathcal{R}_\sigma^{\mathrm{L}}\left(\mathbf{z}_i, \mathbf{c}\right)/\tau_L\right)}{\exp\left(\mathcal{R}_\sigma^{\mathrm{L}}\left(\mathbf{z}_i, \mathbf{c}\right)/\tau_L\right) + \exp\left(\mathcal{R}_\sigma^{\mathrm{L}}\left(\mathbf{z}_j, \mathbf{c}\right)/\tau_L\right)}$$

$\tau_L$ is the temperature parameter. Similarly, with the temperature $\tau_E$, the probability of $R^E$ preferring $\mathbf{z}_i$ over $\mathbf{z}_j$ can be modeled as:

$$Q_{i,j}(i) = \frac{\exp\left(\mathcal{R}^E\left(\mathcal{D}\left(\mathbf{z}_i\right), \mathbf{c}\right)/\tau_E\right)}{\exp\left(\mathcal{R}^E\left(\mathcal{D}\left(\mathbf{z}_i\right), \mathbf{c}\right)/\tau_E\right) + \exp\left(\mathcal{R}^E\left(\mathcal{D}\left(\mathbf{z}_j\right), \mathbf{c}\right)/\tau_E\right)}$$

And thus, we have

$$P_{i,j}^\sigma(m) \propto \exp\left(\mathcal{R}_\sigma^{\mathrm{L}}\left(\mathbf{z}_m, \mathbf{c}\right)/\tau_L\right), \quad Q_{i,j}(m) \propto \exp\left(\mathcal{R}^E\left(\mathcal{D}\left(\mathbf{z}_m\right), \mathbf{c}\right)/\tau_E\right), m \in \{i, j\}.$$

We can construct the KL divergence between the distribution $P_{i,j}^\sigma$ and $Q_{i,j}(i)$ for each $(\mathbf{z}_i, \mathbf{z}_j)$ pair. Our $L_{RM}(\sigma)$ is derived by summing the KL divergence for all three latent pairs as below

$$L_{\mathrm{RM}}(\sigma) = \mathbb{E}_{\mathbf{z}, \mathbf{c}, \omega, n}\left[\sum_{i=0}^{1}\sum_{j=i+1}^{2} D_{\mathrm{KL}}\left(P_{i,j}^\sigma \| \mathtt{stop\_grad}\left(Q_{i,j}\right)\right)\right]. \tag{12}$$

Appendix B.3 includes pseudo-codes for training our RG-LCM with an LRM in Algorithm 4. $L_{\mathrm{RM}}(\sigma)$ also supports matching a $\mathcal{R}^{\mathrm{E}}$ that only output preference over two images. In this case, we can set $\tau_E$ to a small

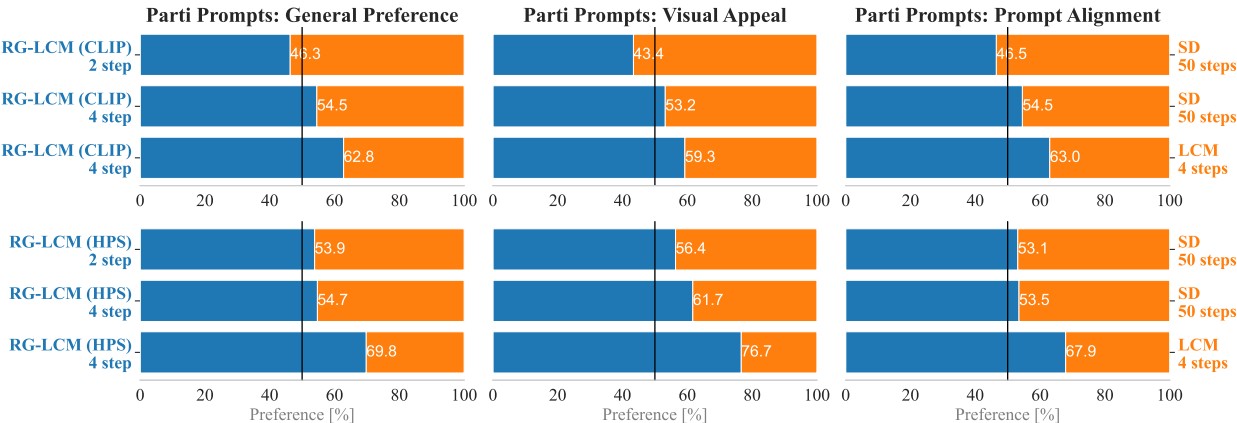

Figure 4: Human evaluation results on the PartiPrompt (1632 prompts) across three evaluation questions. Top row evaluates the RG-LCM (CLIP). Bottom row evaluates the RG-LCM (HPS).

positive number and only give a non-zero positive reward to the sample favored by the expert. Moreover, since $\mathbf{z}_0 = \mathbf{z}$ corresponds to the latent of a real image, we can increase likelihood for $Q_{0,j}$ to prefer $k = 0$.

While calculating $\mathcal{R}^{\mathrm{E}}(\mathcal{D}(\mathbf{z}), \mathbf{c})$ still requires decoding the latent, the application of the `stop_grad` operation eliminates the need for gradient transmission through $\mathcal{D}$, leading to a substantial reduction in memory usage. Moreover, optimizing $\mathcal{R}_\sigma^{\mathrm{L}}$ with $L_{\mathrm{RM}}(\sigma)$ is independent from optimizing $\boldsymbol{f}_\theta$ with $L_{\mathrm{RG\text{-}LCD}}$. Therefore, we can use a smaller batch size to optimize $\mathcal{R}_\sigma^{\mathrm{L}}$ without affecting the batch size used to optimize $\boldsymbol{f}_\theta$.

In essence, our LRM acts as a proxy connecting the LCM $\boldsymbol{f}_\theta$ and the expert RM $\mathcal{R}^{\mathrm{E}}$. As we will show Sec. 5.2, using this indirect feedback from the expert mitigates the issue of reward over-optimization, avoiding high-frequency noise in the generated images.

## 5 Experiment

We perform thorough experiments to demonstrate the effectiveness of our RG-LCD. Sec. 5.1 conducts human evaluation to compare the performance of our methods with baselines. Sec. 5.2 further increases the experiment scales to experiment with a wider array of RMs with automatic metrics. By connecting both evaluation results, we identify problems with the current RMs. Finally, Sec. 5.3 conducts ablation studies on critical design choices.

**Settings** Our training are conducted on the CC12M datasets (Changpinyo et al., 2021), as the LAION-Aesthetics datasets (Schuhmann et al., 2022) used by the original LCM (Luo et al., 2023a) are no longer accessible[1].We distill our LCM from the Stable Diffusion-v2.1 (Rombach et al., 2022) by training for 10K iterations on 8 NVIDIA A100 GPUs without gradient accumulation and set the batch size to reach the maximum capacity of our GPUs. We follow the hyperparameter settings listed in the diffusers (von Platen et al., 2022) library by setting learning rate $1e - 6$, EMA rate $\mu = 0.95$ and the guidance scale range $[\omega_{\min}, \omega_{\max}] = [5, 15]$. As mentioned in Sec. 3.3, we use DDIM (Song et al., 2020a) as our ODE solver $\Psi$ with a skipping step $k = 20$. We include more training details in Appendix A. Appendix D further includes experiment results with diverse teacher T2I models, including Stable Diffusion 1.5 and Stable Diffusion XL.

### 5.1 Evaluating RG-LCD with Human

We train RG-LCM (HPS) and RG-LCM (CLIP) utilizing feedback from HPSv2.1 (Wu et al., 2023a) and CLIPScore (Radford et al., 2021), respectively. CLIPScore evaluates the relevance between text and images, whereas HPSv2.1, derived by fine-tuning CLIPScore with human preference data, is expected to mirror

---

[1] https://laion.ai/notes/laion-maintanence

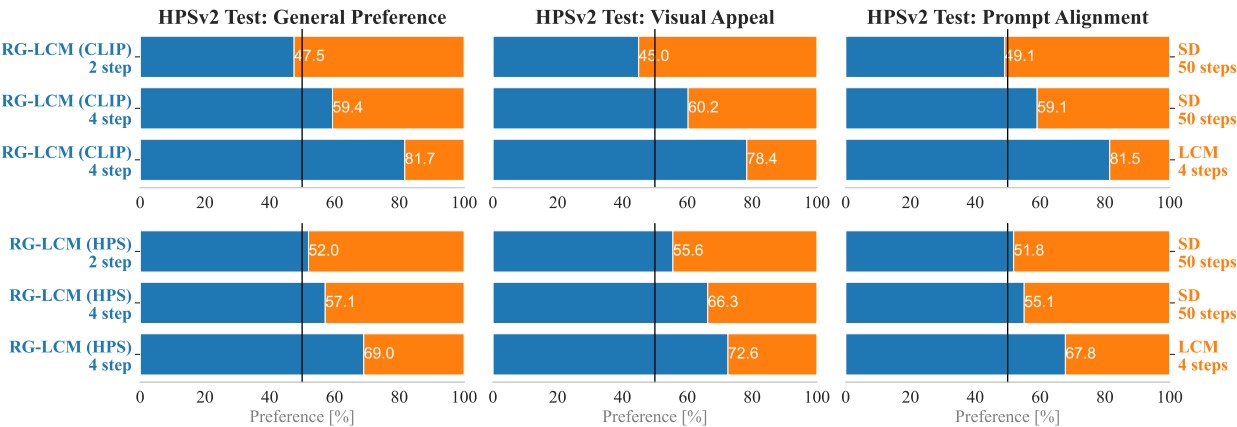

Figure 5: Human evaluation results on the HPSv2 test set (3200 prompts) across three evaluation questions. Top row evaluates the RG-LCM (CLIP). Bottom row evaluates the RG-LCM (HPS).

human preferences more accurately. We choose the teacher LDM (Stable Diffusion v2.1) and a standard LCM distilled from the same teacher LDM as the baseline methods. To demonstrate the efficacy of our methods, we compare the performance of our RG-LCMs over 2-step and 4-step generations against the 50-step generations from the teacher LDM and evaluate the 4-step generation quality of our RG-LCMs against the standard LCM.

We follow a similar evaluation protocol as in (Wallace et al., 2023a) to generate images by conditioning on prompts from Partiprompt (Yu et al., 2022) (1632 prompts) and of HPSv2's test set (Wu et al., 2023a) (3200 prompts). We hire labelers from Amazon Mechanical Turk for a head-to-head comparison of images based on three criteria: Q1 General Preference (Which image do you prefer given the prompt?), Q2 Visual Appeal (Which image is more visually appealing, irrespective of the prompt?), and Q3 Prompt Alignment (Which image better matches the text description?).

The full human evaluation results in Fig. 4 and 5 show that the 2-step generations from RG-LCM (CLIP) are generally preferred (Q1) over the 50-step generations of the teacher LDM in both prompt sets, representing a 25-fold acceleration in inference speed. Even with CLIPScore feedback, the 4-step generations from our RG-LCM are generally preferred (Q1) over the baseline methods. This indicates a noteworthy achievement, given that CLIPScore does not train on human preference data. Surprisingly, on the HPSv2 prompt set, the 4-step generations from the RG-LCM (CLIP) are more preferred (59.4% against 50-step DDIM samples from SD and 81.7% against 4-step LCM samples) compared to the 4-step generations of the RG-LCM (CLIP) (57.1% against 50-step DDIM samples from SD, and 69.0% against 4-step LCM samples).

To investigate this phenomenon, we observe that both RG-LCMs score similarly in General Preference (Q1) and Prompt Alignment (Q3). However, the RG-LCM (CLIP) is rated slightly lower in Visual Appeal (Q2) than in the other criteria, whereas the RG-LCM (HPS) is rated significantly higher for Q2 compared to Q1 and Q3. This distinction highlights that CLIPScore's primary contribution is enhancing text-image alignment, whereas an RM like HPSv2.1 particularly focuses on improving visual quality. Thus, when over-optimizing towards HPSv2.1, the RG-LCM (HPS) can be biased in generating visually appealing samples by sacrificing prompt alignment.

## 5.2 Evaluating RG-LCD with Automatic Metrics

In this section, we further train RG-LCD (ImgRwd) and RG-LCD (Pick) by leveraging feedback from ImageReward (Xu et al., 2024) and PickScore (Kirstain et al., 2024). Both of these RMs are trained on human preference data. We will use automatic metrics to perform a large-scale evaluation of the performance of different models. As we have human evaluation results for RG-LCD (HPS) and RG-LCD (CLIP), we can

| Models | NFEs | Human Preference Score v2.1 ↑ | | | | FID-30K ↓ |
| --- | --- | --- | --- | --- | --- | --- |
| | | Anime | Photo | Concept-Art | Paintings | MS-COCO |
| LCM | 4 | 22.40 | 19.17 | 18.86 | 20.55 | 19.05 |
| Stable Diffusion v2.1 | 50 | 25.66 | 24.37 | 24.58 | 25.72 | **12.66** |
| RG-LCM (CLIP) | 2 | 26.32 | 25.01 | 25.27 | 26.71 | 18.06 |
| RG-LCM (CLIP) | 4 | 27.80 | 26.92 | 27.04 | 28.11 | 19.22 |
| RG-LCM (Pick) | 2 | 26.44 | 28.26 | 28.24 | 29.04 | 22.84 |
| RG-LCM (Pick) | 4 | 27.33 | 29.42 | 29.29 | 30.26 | 22.02 |
| RG-LCM (Pick) + LRM | 2 | 23.82 | 21.31 | 21.90 | 22.99 | 15.91 |
| RG-LCM (Pick) + LRM | 4 | 25.17 | 23.06 | 22.90 | 24.87 | 16.27 |
| RG-LCM (ImgRwd) | 2 | 29.65 | 31.03 | 31.15 | 32.00 | 32.12 |
| RG-LCM (ImgRwd) | 4 | 30.26 | 31.83 | 31.88 | 32.73 | 42.69 |
| RG-LCM (ImgRwd) + LRM | 2 | 25.64 | 25.61 | 25.82 | 25.75 | 17.57 |
| RG-LCM (ImgRwd) + LRM | 4 | 26.84 | 26.72 | 26.72 | 27.30 | 17.20 |
| RG-LCM (HPS) | 2 | 30.85 | 33.66 | 33.35 | 33.66 | 24.04 |
| RG-LCM (HPS) | 4 | **31.83** | **34.84** | **34.43** | **34.75** | 25.11 |
| RG-LCM (HPS) + LRM | 2 | 27.58 | 25.94 | 26.77 | 27.24 | 16.71 |
| RG-LCM (HPS) + LRM | 4 | 28.53 | 27.49 | 27.94 | 28.87 | 17.52 |

Table 1: Evaluation of our RG-LCMs on the HPSv2 test prompts and MS-COCO datasets. NFEs denote the number of function evaluations during inference. We train RG-LCMs with CLIPScore, PickScore, ImageReward (ImgRwd) and HPSv2.1. We employ the HPSv2.1 to evaluate the generations on the HPSv2 Benchmark's test set. We calculate the FID of the generations on the MS-COCO. Except trained with CLIPScore, our RG-LCMs achieve better HPSv2.1 scores on HPSv2 test prompts at the expense of higher FIDs on MS-COCO. Integrating a LRM into our RG-LCD process allows for simultaneous improvement on HPSv2.1 scores on HPSV2 test prompts and FID on MS-COCO against the baseline LCM.

also evaluate the quality of the automatic metrics. For each RG-LCD, we collect their 2-step and 4-step generations by conditioning on prompts from HPSv2's test set and measuring the HPSv2.1 score associated with the samples. To comprehensively understand the sample quality from different models, we further generate images conditioned on the prompts of MS-COCO (Lin et al., 2014) and measure their Fréchet Inception Distance (FID) to the ground truth images.

Table 1 presents the full evaluation results with the automatic metrics. Except for RG-LCM (CLIP), all the other RG-LCMs achieve higher HPSv2.1 scores than the baseline LCM but at the expense of higher FID values on the MS-COCO dataset. Specifically, the RG-LCM (ImgRwd) model exhibits a notably high FID value, yet it still secures an impressive HPSv2.1 score when evaluated on HPSv2 test prompts. The elevated FID value aligns with expectations, as Figure 3 illustrates that optimization directed towards ImageReward tends to introduce a significant amount of high-frequency noise into the generated images. Surprisingly, these high-frequency noises do not adversely affect the HPSv2.1 scores. Furthermore, the HPSv2.1 scores do not capture the human preference for the 4-step samples from RG-LCM (CLIP) by giving the highest score to RG-LCM (HPS)'s 4-step samples, contrary to what is depicted in human evaluation shown in Fig. 5.

These observations suggest that the HPSv2.1 score, as a metric, has limitations and requires further refinement. We conjecture that the *Resize* operation, which happens during the preprocessing phase, causes the HPSv2.1 model to overlook the high-frequency noise during reward calculation. As illustrated in Fig. 3, the high-frequency noise becomes less perceptible when images are reduced in size. Although resizing operations enhance efficiency in tasks such as image classification (Lu & Weng, 2007; Deng et al., 2009; He et al., 2016) and facilitate high-level text-image understanding (Radford et al., 2021), they prevent the model from capturing critical visual nuances that are vital for accurately reflecting human preferences. Consequently, we advocate for future RMs to **exclude the *Resize* operation**. One potential approach could involve training an LRM, as in our paper, to learn human preferences in the latent space without resizing input images.

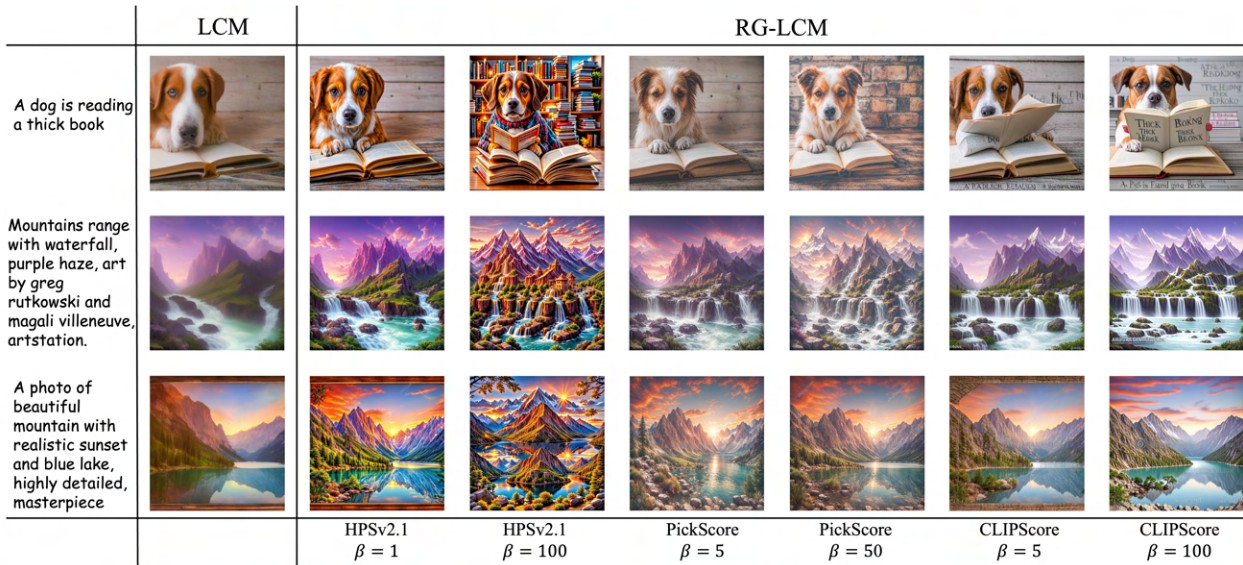

Figure 6: Ablating the reward scale $\beta$ for different reward functions. All samples are generated with 4 steps. We observe that over-optimizing RMs trained with preference data prioritizes visual appeal over text alignment, whereas over-optimizing CLIPScore compromises visual attractiveness in favor of text alignment.

Connecting Table 1 with the human evaluation results in Fig. 5 suggests that images that achieve a high HPSv2.1 score and a low FID on MS-COCO are more aligned with human preferences. Moreover, this desirable outcome can be accomplished by integrating an LRM into our RG-LCD. Although these correlations do not imply causality, they underscore the potential benefits of utilizing an LRM in the RG-LCD process. As depicted in the bottom row in Fig. 3, the images generated by RG-LCM (ImgRwd) that integrates an LRM do not suffer from high-frequency noise, contributing to their improved FID on MS-COCO. In Appendix C, we include additional samples for each RG-LCM in Table 1.

### 5.3  Ablation Study

**Ablation on the reward scale $\beta$.** We use the hyperparameter $\beta$ to determine the optimization strength towards the RM. We are especially interested in the impact of an extremely large $\beta$, which can lead to reward over-optimization (Kim et al., 2023b). We already know that over-optimizing the ImageReward can lead to the introduction of high-frequency noise in the generated images. To expand our understanding, we conduct experiments a wider array of RMs including HPSv2.1, PickScore and CLIPScore and evaluate whether over-optimizing these RMs will also leads to similar high-frequency noise.

The results in Figure 6 reveal that an extremely large $\beta$ value does not introduce the high-frequency noise when using HPSv2.1, PickScore, and CLIPScore, even though all these metrics resize input images to 224x224 pixels as in ImageReward. Notably, over-optimization of HPSv2.1 leads to generating images with repetitive objects as described in the text prompts and increases color saturation. Conversely, over-optimization of PickScore tends to result in images with more muted colors. On the other hand, excessive optimization of CLIPScore results in images where the text prompts are visibly incorporated into the imagery. These findings align with the discussions in Sec. 5.1, suggesting that optimizing towards a preference-trained RM generally prioritizes visual appeal over text alignment. In contrast, over-optimizing CLIPScore compromises visual attractiveness in favor of text alignment. We include additional image samples in Appendix C.

**Ablation on the training iterations.** In total, we train each model for 10K iterations. We take checkpoints from 1K, 2K, 4K, and 10K iterations and sample images with the same prompts and seeds. We can observe performing RG-LCD with RMs that facilitate the visual appeal of the generated images also results

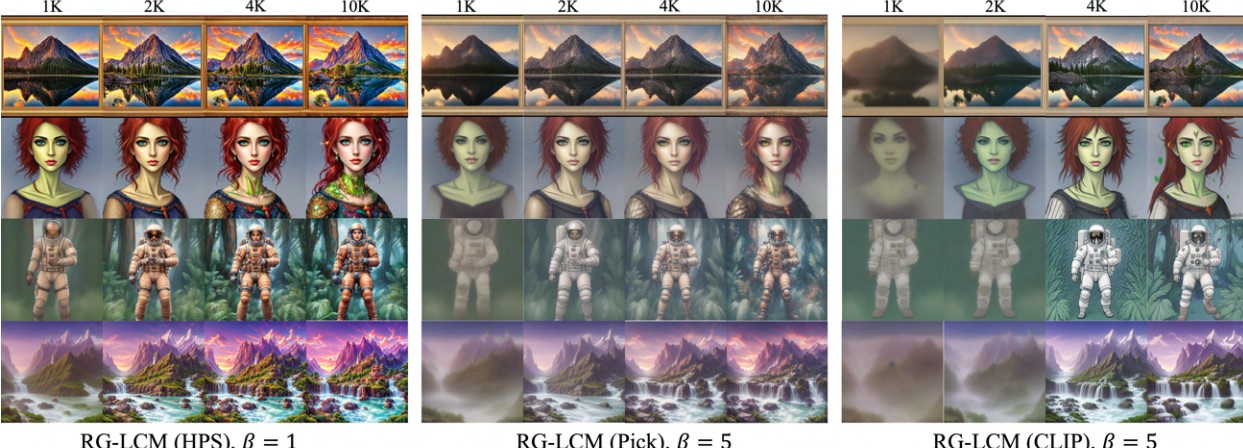

Figure 7: Ablation study on the number of training iterations. We generate all samples with 4 steps. We observe that RG-LCM, which learns from an RM that prioritizes visual appeal, can generate high-quality images with fewer training iterations.

in fast training, as the checkpoint at the 2K iterations can already produce high-quality images. In contrast, the images generated by RG-LCM (CLIP) still generate blurry images after training for 2K iterations.

## 6  Conclusion

In this paper, we introduce RG-LCD, a novel strategy that integrates feedback from an RM into the LCD process. The RG-LCM learned via our method enjoys better sample quality while facilitating fast inference, benefiting from additional computational resources allocated to align with human preferences. By evaluating using prompts from the HPSv2 (Wu et al., 2023a) test set and PartiPrompt (Yu et al., 2022), we empirically show that humans favor the 2-step generations of our RG-LCD (HPS) over the 50-step DDIM generations of the teacher LDM. This represents a 25-fold increase in terms of inference speed without a loss in quality. Moreover, even when using CLIPScore—a model not fine-tuned on human preferences—our method's 4-step generations still surpass the 50-step DDIM generations from the teacher LDM.

We also identify that directly optimizing towards an imperfect RM, e.g., ImageReward, can cause high-frequency noise in generated images. To reconcile the issue, we propose integrating an LRM into the RG-LCD framework. Notably, our methods not only prevents reward over-optimization but also avoids passing gradients through the VAE decoder and facilitates learning from non-differentiable RMs.

## 7  Limitation and Impact Statement

While our RG-LCD marks a critical advancement in the realm of efficient text-to-image synthesis, introducing an acceleration in the generation process without compromising on image quality, it is important to recognize certain limitations. The approach relies on employing a reward model that reflects human preference, which, while effective in improving image quality metrics, may introduce additional costs in the training pipeline and necessitate fine-tuning to adapt to various domains or datasets. Despite these challenges, the impact of RG-LCD is profound, offering a scalable solution that significantly enhances the accessibility and practicality of generating high-fidelity images at a remarkable speed. This innovation not only broadens the potential applications in fields ranging from digital art to visual content creation but also sets a new benchmark for future research in text-to-image synthesis, emphasizing the importance of human-centric design in the development of generative AI technologies.

## Acknowledgement

The work was funded by an unrestricted gift from Google. We would like to thank Google for their generous sponsorship. The views and conclusions contained in this document are those of the authors and should not be interpreted as representing the sponsors' official policy, expressed or inferred.

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

# Appendix

In the main paper, we distill our RG-LCD from the Stable Diffusion-v2.1 (768 x 768) (Rombach et al., 2022). Fig. 8 further shows the samples from our RG-LCM (HPSv) distilled from Stable Diffusion-v2.1-base (512 x 512) (Rombach et al., 2022). The rest of the appendix is structured as below

- Appendix A details the experimental setup and hyperparameter configurations.

- Appendix B elaborates on the training processes and sampling techniques from a (latent) CM.

- Appendix C shows extra samples generated by various models.

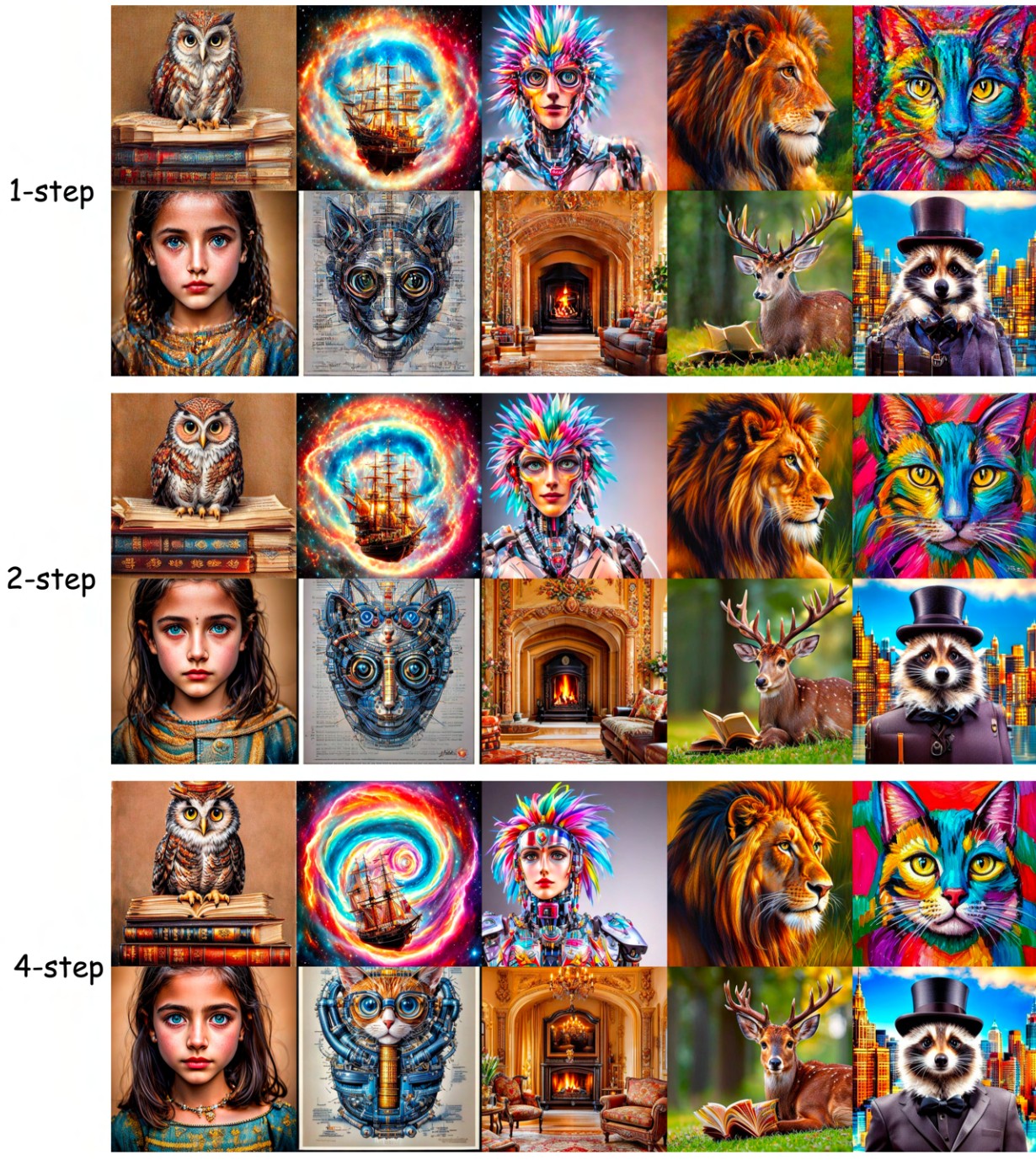

Figure 8: Samples from our RG-LCM (HPSv2.1) with the teacher Stable Diffusion v2.1-base. The resolution is 512 x 512.

|   | CLIPScore | PickScore | ImageReward | HPSv2.1 |
|---|-----------|-----------|-------------|---------|
| $\beta$ | 5.0 | 5.0 | 1.0 | 1.0 |

Table 2: $\beta$ for different RG-LCMs when training with different RMs.

# A  Additional Experimental Details and Hyperparameters (HPs)

For qualitative evaluation, we ensure consistency across all methods by using the same random seed for head-to-head image comparisons.

As mentioned in Sec. 5, our training are conducted on the CC12M datasets (Changpinyo et al., 2021), as the LAION-Aesthetics datasets (Schuhmann et al., 2022) used in the original LCM paper (Luo et al., 2023a) are not accessible[2]. We train all LCMs (including RG-LCMs and the standard LCM) by distilling from the teacher LDM Stable Diffusion-v2.1 (Rombach et al., 2022) for 10K gradient steps on 8 NVIDIA A100 GPUs. When learning the standard LCM, we use a batch size 32 on each GPU (256 effective batch size). For RG-LCMs, we use a batch size 5 on each GPU (40 effective batch size). Interestingly, we observe that different batch sizes do not impact the final performance too much.

We use the same set of hyperparameters (HP) for training RG-LCM and the standard LCM by following the settings listed in the diffusers (von Platen et al., 2022) library, except that RG-LCM has a unique HP $\beta$. Specifically, we set the learning rate $1e-6$, EMA rate $\mu = 0.95$ and the guidance scale range $[\omega_{\min}, \omega_{\max}] = [5, 15]$. We include more training details in Appendix A. Following the practice in (Luo et al., 2023a), we initialize $\boldsymbol{f}_\theta$ with the same parameters as the teacher LDM. We further encode the CFG scale $\omega$ by applying the Fourier embedding to $\omega$ and integrate it into the LCM backbone by adding the projected $\omega$-embedding into the original embedding as in (Meng et al., 2023).

As mentioned in Sec. 3.3, we use DDIM (Song et al., 2020a) as our ODE solver $\Psi$ with a skipping step $k = 20$, the formula of the DDIM ODE solver $\Psi_{\mathrm{DDIM}}$ from $t_{n+k}$ to $t_n$ is given below (Luo et al., 2023a)

$$\Psi_{\mathrm{DDIM}}\left(\boldsymbol{z}_{t_{n+k}}, t_{n+k}, t_n, \boldsymbol{c}\right) = \underbrace{\frac{\alpha_{t_n}}{\alpha_{t_{n+k}}} \boldsymbol{z}_{t_{n+k}} - \beta_{t_n} \left(\frac{\beta_{t_{n+k}} \cdot \alpha_{t_n}}{\alpha_{t_{n+k}} \cdot \beta_{t_n}} - 1\right) \hat{\boldsymbol{\epsilon}}_\psi \left(\boldsymbol{z}_{t_{n+k}}, \boldsymbol{c}, t_{n+k}\right)}_{\text{DDIM Estimated } \boldsymbol{z}_{t_n}} - \boldsymbol{z}_{t_{n+k}}, \qquad (13)$$

where $\hat{\boldsymbol{\epsilon}}_\psi$ denotes the noise prediction model from the teacher LDM. $\alpha_{t_n}$ and $\beta_{t_n}$ specify the noise schedule. For the forward process SDE defined in equation 1, we have

$$\boldsymbol{\mu}(t) = \frac{\mathrm{d}\log\alpha(t)}{\mathrm{d}t}, \quad \sigma^2(t) = \frac{\mathrm{d}\beta^2(t)}{\mathrm{d}t} - 2\frac{\mathrm{d}\log\alpha(t)}{\mathrm{d}t}\beta^2(t). \qquad (14)$$

As a result, we have $p_t(\mathbf{x}_t) = \mathcal{N}(\mathbf{x}_t|\alpha(t), \beta^2(t)\mathbf{I})$. We refer interested readers to the original LCM paper (Luo et al., 2023a) for further details.

**Reward scale $\beta$ for different RG-LCMs with different RMs**. In Sec. 5.1 and 5.2, we train our RG-LCMs with different RMs, including CLIPScore (Radford et al., 2021), PickScore (Kirstain et al., 2024), ImageReward (Xu et al., 2024) and HPSv2.1 (Wu et al., 2023a). Table 2 shows the $\beta$ we used for different RMs when obtaining the results in Fig. 4, 5 and Table 1.

**Details for integrating an LRM into RG-LCM** As discussed in Sec. 4.2, the LRM admits a similar architecture as the CLIP (Radford et al., 2021) model, with the distinction of replacing the visual encoder with a latent encoder. We retain the original pretrained text encoder and focus on pretraining the latent encoder from scratch. This process mirrors the CLIP's pretraining approach, minimizing the same contrastive loss on the CC12M datasets (Changpinyo et al., 2021). The image latent is extracted with the same VAE encoder used in the teacher LDM Stable Diffusion-v2.1. Upon completing the pretraining phase, the LRM demonstrates promising initial results, achieving a zero-shot Top-1 Accuracy of 38.8% and Top-5 Accuracy of

---

[2]https://laion.ai/notes/laion-maintanence

66.47% on the ImageNet validation set (Deng et al., 2009). These metrics underscore the model's fundamental capability in understanding text-image alignments.

During the RG-LCD process, we finetune the LRM to match the preference of an expert RM. We train the last 2 layer of the latent encoder and the last 5 layers of the text encoder. We set the learning rate to 0.0000033 following (Wu et al., 2023a). Note that we do not perform heavy HP searches to determine their optimal values.

As we are finetuning our LRM, there is a potential risk of overfitting the model to the training datasets, which could degrade the quality of generated outputs if training continues indefinitely. We emphasize that this is unlikely to pose a problem for our method. In practice, we use a large and diverse text-image dataset, such as CC12M. We also fixed the training to 10K iterations and observed stable performance without encountering any training instability. We hypothesize that performance degradation would only occur if training exceeds one full epoch of the dataset. However, even with a large batch size of 256, one epoch would require 12M / 256 = 47.9K iterations, which is far beyond the 10K iterations we used. Therefore, early stopping is not a critical concern for our approach.

Nonetheless, we could still implement a stopping criterion by monitoring the average rewards of training batches. We can stop the LRM training when the average rewards converge to a specific value, ensuring the LRM is not overtrained.

# B   Training and Sampling from (Latent) CM

## B.1   Multistep sampling from a learned CM and LCM

We provide the pseudo-codes for multistep consistency sampling (Song et al., 2023) and multistep latent consistency sampling (Luo et al., 2023a) procedures in Algorithm 1 and Algorithm 2, respectively. The multistep sampling procedures alternate between the consistency mapping and noise-injection steps, trading additional computation resources for better sample quality. In the $n$-th iteration, we first perturb the predicted sample $\mathbf{x}$ (or $\mathbf{z}$) with Gaussian noise to obtain $\hat{\mathbf{x}}_{\tau_n}$ (or $\hat{\mathbf{z}}_{\tau_n}$). We then map the noisy sample $\hat{\mathbf{x}}_{\tau_n}$ (or $\hat{\mathbf{z}}_{\tau_n}$) to obtain a new $\mathbf{x}$ (or $\mathbf{z}$).

---

**Algorithm 1** Multistep Consistency Sampling

**Require:** CM $\boldsymbol{f}_\theta$, steps $N$, timestep sequence $\tau_1 > \tau_2 > \cdots > \tau_{N-1}$, noise schedule $\alpha(t), \beta(t)$.
    Sample initial noise $\hat{\mathbf{x}}_T \sim \mathcal{N}(\mathbf{0}, \mathbf{I})$
    $\mathbf{x} \leftarrow \boldsymbol{f}_\theta(\hat{\mathbf{x}}_T, T)$
    **for** $n = 1, \ldots, N-1$ **do**
        Sample $\hat{\mathbf{x}}_{\tau_n} \sim \mathcal{N}(\alpha(\tau_n)\mathbf{x}, \beta^2(\tau_n)\mathbf{I})$
        $\mathbf{x} \leftarrow \boldsymbol{f}(\hat{\mathbf{x}}_{\tau_n}, \tau_n)$
    **end for**
    **Return x**

---

**Algorithm 2** Multistep Latent Consistency Sampling

**Require:** LCM $\boldsymbol{f}_\theta$, steps $N$, timestep sequence $\tau_1 > \tau_2 > \cdots > \tau_{N-1}$, noise schedule $\alpha(t), \beta(t)$, text prompt $\mathbf{c}$, CFG scale $\omega$, VAE decoder $\mathcal{D}$.
    Sample initial noise $\hat{\mathbf{z}}_T \sim \mathcal{N}(\mathbf{0}, \mathbf{I})$
    $\mathbf{z} \leftarrow \boldsymbol{f}_\theta(\hat{\mathbf{z}}_T, \omega, \mathbf{c}, T)$
    **for** $n = 1, \ldots, N-1$ **do**
        Sample $\hat{\mathbf{z}}_{\tau_n} \sim \mathcal{N}(\alpha(\tau_n)\mathbf{z}, \beta^2(\tau_n)\mathbf{I})$
        $\mathbf{z} \leftarrow \boldsymbol{f}_\theta(\hat{\mathbf{z}}_{\tau_n}, \omega, \mathbf{c}, T)$
    **end for**
    **Return** $\mathcal{D}(\mathbf{z})$

---

## B.2   Training procedures of RG-LCD

---

**Algorithm 3** Reward Guided Latent Consistency Distillation

**Require:** dataset $\mathcal{D}$, initial model parameter $\theta$, learning rate $\eta$, ODE solver $\Psi$, distance metric $d$, EMA rate $\mu$, learning rate $\eta$, noise schedule $\alpha(t), \beta(t)$, guidance scale $[\omega_{\min}, \omega_{\max}]$, skipping interval $k$, VAE encoder $\mathcal{E}$, decoder $\mathcal{D}$, reward model $\mathcal{R}$, reward scale $\beta$
    Encoding training data into latent space: $\mathcal{D}_z = \{(\boldsymbol{z}, \boldsymbol{c}) \mid \boldsymbol{z} = E(\boldsymbol{x}), (\boldsymbol{x}, \boldsymbol{c}) \in \mathcal{D}\}$
    $\theta^- \leftarrow \theta$
    **repeat**
        Sample $(\boldsymbol{z}, \boldsymbol{c}) \sim \mathcal{D}_z, n \sim \mathcal{U}[1, N-k]$ and $\omega \sim [\omega_{\min}, \omega_{\max}]$
        Sample $\boldsymbol{z}_{t_{n+k}} \sim \mathcal{N}\left(\alpha\left(t_{n+k}\right)\boldsymbol{z}; \sigma^2\left(t_{n+k}\right)\mathbf{I}\right)$
        $\hat{\boldsymbol{z}}_{t_n}^{\Psi,\omega} \leftarrow \boldsymbol{z}_{t_{n+k}} + (1+\omega)\Psi\left(\boldsymbol{z}_{t_{n+k}}, t_{n+k}, t_n, \boldsymbol{c}\right) - \omega\Psi\left(\boldsymbol{z}_{t_{n+k}}, t_{n+k}, t_n, \varnothing\right)$
        $\mathcal{L}\left(\theta, \theta^-; \Psi\right) \leftarrow d\left(\boldsymbol{f}_\theta\left(\boldsymbol{z}_{t_{n+k}}, \omega, \boldsymbol{c}, t_{n+k}\right), \boldsymbol{f}_{\theta^-}\left(\hat{\boldsymbol{z}}_{t_n}^{\Psi,\omega}, \omega, \boldsymbol{c}, t_n\right)\right)$ $- \beta \cdot \mathcal{R}\left(\mathcal{D}\left(\boldsymbol{f}_\theta\left(\mathbf{z}_{t_{n+k}}, \omega, \mathbf{c}, t_{n+k}\right)\right), \mathbf{c}\right)$
        $\theta \leftarrow \theta - \eta\nabla_\theta\mathcal{L}\left(\theta, \theta^-\right)$
        $\theta^- \leftarrow \texttt{stop\_grad}\left(\mu\theta^- + (1-\mu)\theta\right)$
    **until** convergence

---

Algorithm 3 presents the pseudo-codes for our RG-LCD. We use the red color to highlight the difference between our RG-LCD and the standard LCD (Luo et al., 2023a).

## B.3   Training procedures of RG-LCD with a Latent Proxy RM

Algorithm 4 presents the training codes for our RG-LCD with a Latent Proxy RM.

---

**Algorithm 4** Reward Guided Latent Consistency Distillation with a Latent Proxy RM

---

**Require:** dataset $\mathcal{D}$, initial model parameter $\theta$, learning rate $\eta$, ODE solver $\Psi$, distance metric $d$, EMA rate $\mu$, learning rates $\eta_1$, $\eta_2$, noise schedule $\alpha(t), \beta(t)$, guidance scale $[\omega_{\min}, \omega_{\max}]$, skipping interval $k$, VAE encoder $\mathcal{E}$, decoder $\mathcal{D}$, reward scale $\beta$, expert RM $\mathcal{R}^E$, LRM $\mathcal{R}^L_\sigma$, temperature $\tau_E$ and $\tau_L$
 Encoding training data into latent space: $\mathcal{D}_z = \{(\boldsymbol{z}, \boldsymbol{c}) \mid \boldsymbol{z} = E(\boldsymbol{x}), (\boldsymbol{x}, \boldsymbol{c}) \in \mathcal{D}\}$
 $\theta^- \leftarrow \theta$
 **repeat**
    # Calculate the Training loss for $\boldsymbol{f}_\theta$
    Sample $(\boldsymbol{z}, \boldsymbol{c}) \sim \mathcal{D}_z, n \sim \mathcal{U}[1, N-k]$ and $\omega \sim [\omega_{\min}, \omega_{\max}]$
    Sample $\boldsymbol{z}_{t_{n+k}} \sim \mathcal{N}\left(\alpha\left(t_{n+k}\right)\boldsymbol{z}; \sigma^2\left(t_{n+k}\right)\mathbf{I}\right)$
    $\hat{\boldsymbol{z}}^{\Psi,\omega}_{t_n} \leftarrow \boldsymbol{z}_{t_{n+k}} + (1+\omega)\Psi\left(\boldsymbol{z}_{t_{n+k}}, t_{n+k}, t_n, \boldsymbol{c}\right) - \omega\Psi\left(\boldsymbol{z}_{t_{n+k}}, t_{n+k}, t_n, \varnothing\right)$
    Detach the parameter $\sigma$ of $\mathcal{R}^L_\sigma$
    $\mathcal{L}\left(\theta, \theta^-; \Psi, \sigma\right) \leftarrow d\left(\boldsymbol{f}_\theta\left(\boldsymbol{z}_{t_{n+k}}, \omega, \boldsymbol{c}, t_{n+k}\right), \boldsymbol{f}_{\theta^-}\left(\hat{\boldsymbol{z}}^{\Psi,\omega}_{t_n}, \omega, \boldsymbol{c}, t_n\right)\right) - \beta \cdot \mathcal{R}^L_\sigma\left(\boldsymbol{z}_{t_{n+k}}, \mathbf{c}\right)$
    # Calculate the Training loss for $\mathcal{R}^E_\sigma$
    $\mathbf{z}_0 \leftarrow \mathbf{z}, \mathbf{z}_1 \leftarrow \boldsymbol{f}_\theta\left(\boldsymbol{z}_{t_{n+k}}, \omega, \boldsymbol{c}, t_{n+k}\right), \mathbf{z}_2 \leftarrow \boldsymbol{f}_{\theta^-}\left(\hat{\boldsymbol{z}}^{\Psi,\omega}_{t_n}, \omega, \boldsymbol{c}, t_n\right)$
    **for** $i \in \{0, 1\}$ **do**
        **for** $j \in \{i, 2\}$ **do**
            Derive the preference distribution: $P^\sigma_{i,j}(m) \propto \exp\left(\mathcal{R}^L_\sigma\left(\mathbf{z}_m, \mathbf{c}\right)/\tau_L\right), \quad m \in \{i, j\}$
            Derive the preference distribution: $Q_{i,j}(m) \propto \exp\left(\mathcal{R}^E\left(\mathcal{D}\left(\mathbf{z}_m\right), \mathbf{c}\right)/\tau_E\right), \quad m \in \{i, j\}$
        **end for**
    **end for**
    $L_{\text{RM}}(\sigma) \leftarrow \sum^1_{i=0}\sum^2_{j=i+1} D_{\text{KL}}\left(P^\sigma_{i,j} \| \texttt{stop\_grad}\left(Q_{i,j}\right)\right)$
    # Update the learnable parameters via gradient descent
    $\theta \leftarrow \theta - \eta_1 \nabla_\theta \mathcal{L}\left(\theta, \theta^-\right)$
    $\theta^- \leftarrow \texttt{stop\_grad}(\mu\theta^- + (1-\mu)\theta)$
    $\sigma \leftarrow \theta - \eta_2 \nabla_\sigma L_{\text{RM}}(\sigma)$
 **until** convergence

---

## C   Additional Qualitative Results

We provide additional samples from our RG-LCMs with different RMs compared with the baseline LCM and teacher Stable Diffusion v2.1 in Fig. 9 and 10.

The prompts for images in Fig. 9 in the left-to-right order are given below

- Van Gogh painting of a teacup on the desk

- Impressionist painting of a cat, textured, hypermodern

- photo of a kid playing , snow filling the air

- A fluffy owl sits atop a stack of antique books in a detailed and moody illustration.

- a deer reading a book

- a photo of a monkey wearing glasses in a suit

The prompts for images in Fig. 10 in the left-to-right order are given below

- ornate archway inset with matching fireplace in room

- Poster of a mechanical cat, techical Schematics viewed from front

- portrait of a person with Cthulhu features, painted by Bouguereau.

- a serene nighttime cityscape with lake reflections, fruit trees

- Teddy bears working on new AI research on the moon in the 1980s.

- A cinematic shot of robot with colorful feathers

Fig. 11 presents image samples when integrating a latent proxy RM (LRM) into our RG-LCD procedures. The prompts in the left-to-right order are given below

- a man in a brown blazer standing in front of smoke, backlit, in the style of gritty hollywood glamour, light brown and emerald, movie still, emphasis on facial expression, robert bevan, violent, dappled

- a cute pokemon resembling a blue duck wearing a puffy coat

- Highly detailed photograph of a meal with many dishes.

Fig. 12 further qualitatively compares RG-LCM (HPS) + LRM and the standard LCM. The prompts in the top-to-bottom order are given below

- (Masterpiece:1. 5), RAW photo, film grain, (best quality:1. 5), (photorealistic), realistic, real picture, intricate details, photo of full body a cute cat in a medieval warrior costume, ((wastelands background)), diamond crown on head, (((dark background)))

- back view of a woman walking at Shibuya Tokyo, shot on Afga Vista 400, night with neon side lighting

- Fall And Autumn Wallpaper Daniel Wall Rainy Day In Autumn Painting Oil Artwork

- A bird-eye shot photograph of New York City, shot on Lomography Color Negative 800

- painting of forest and pond

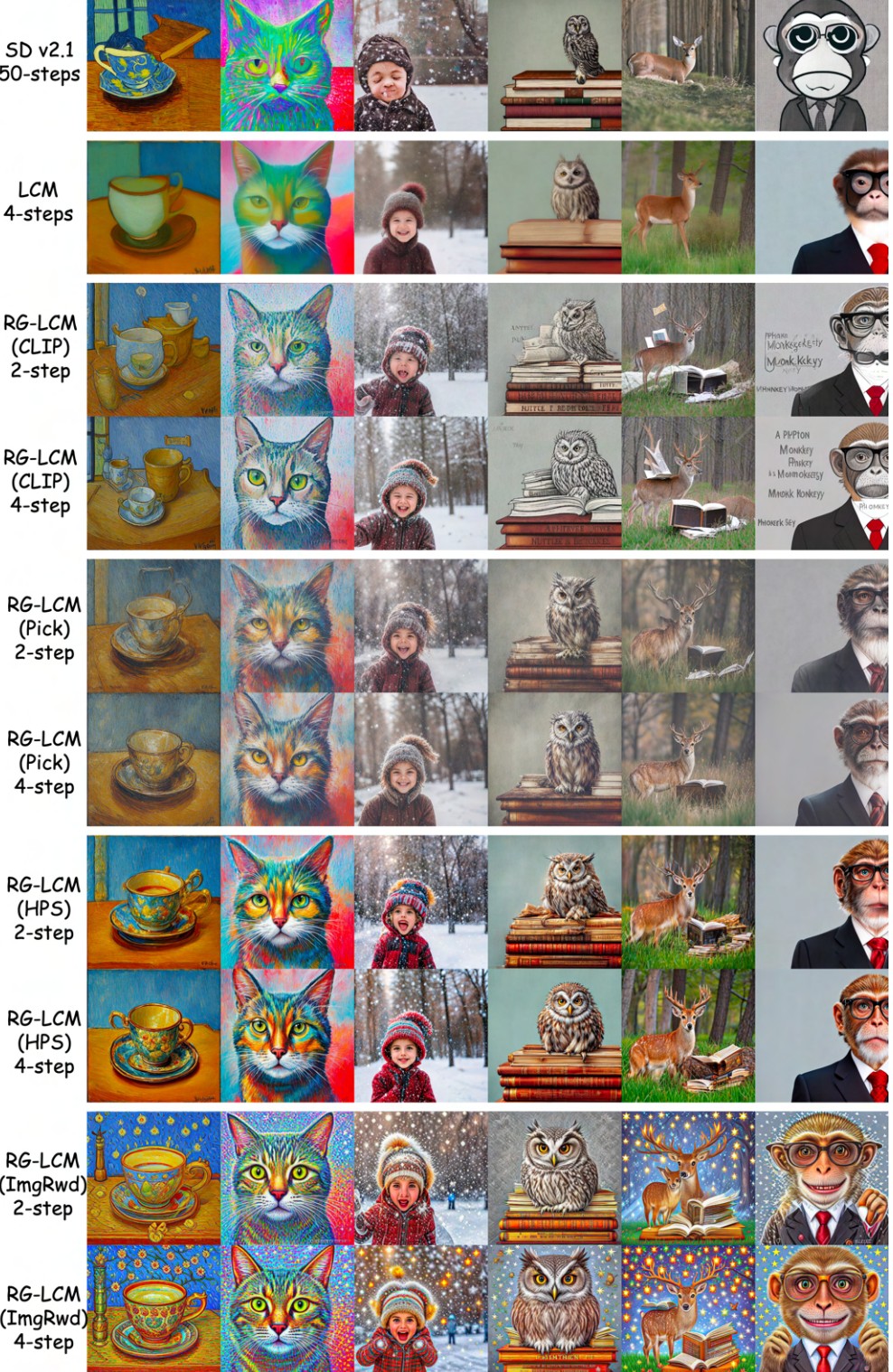

Figure 9: Samples from our RG-LCMs with different RMs compared with the baseline LCM and teacher Stable Diffusion v2.1.

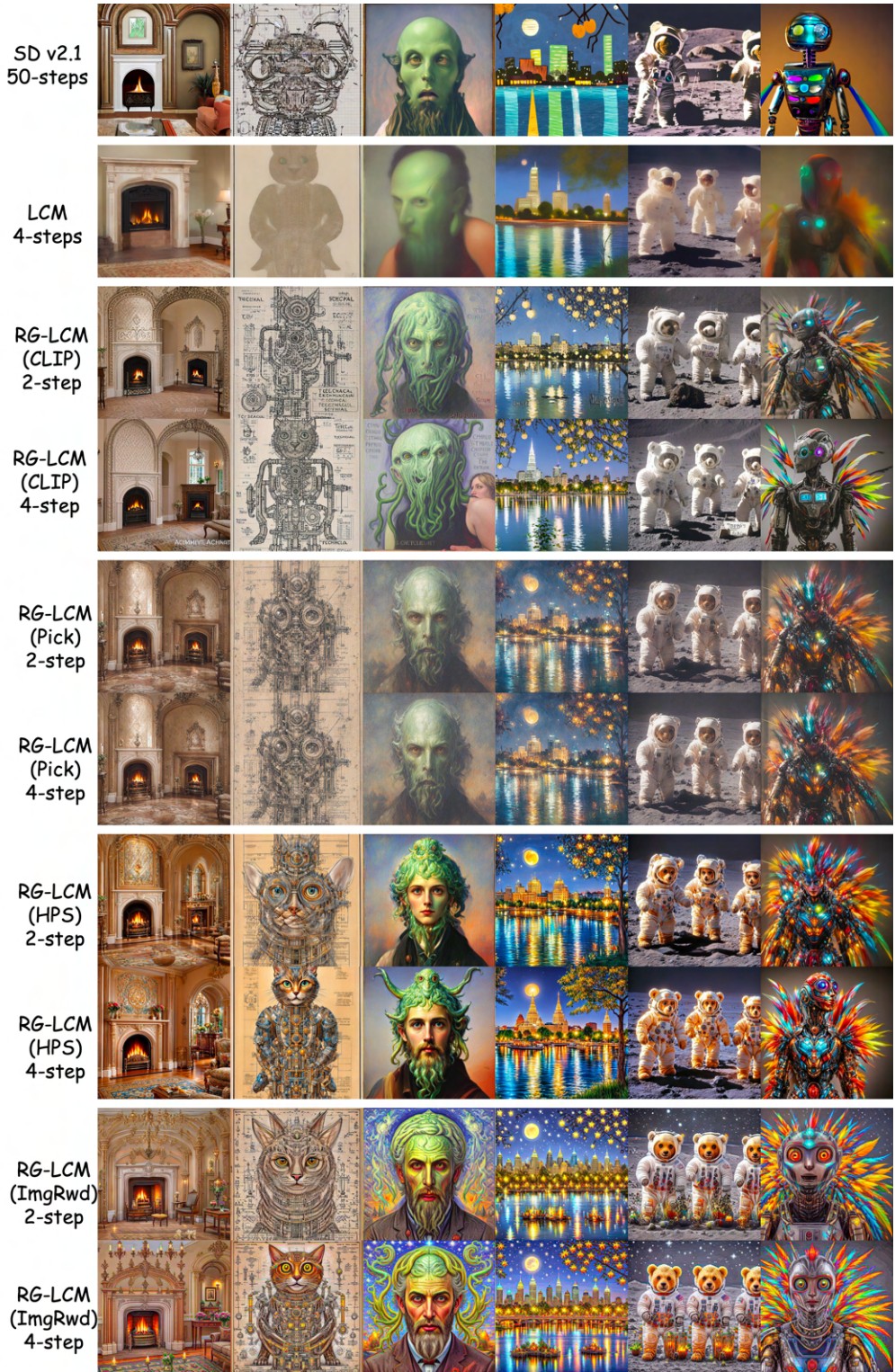

Figure 10: More samples from our RG-LCMs with different RMs compared with the baseline LCM and teacher Stable Diffusion v2.1.

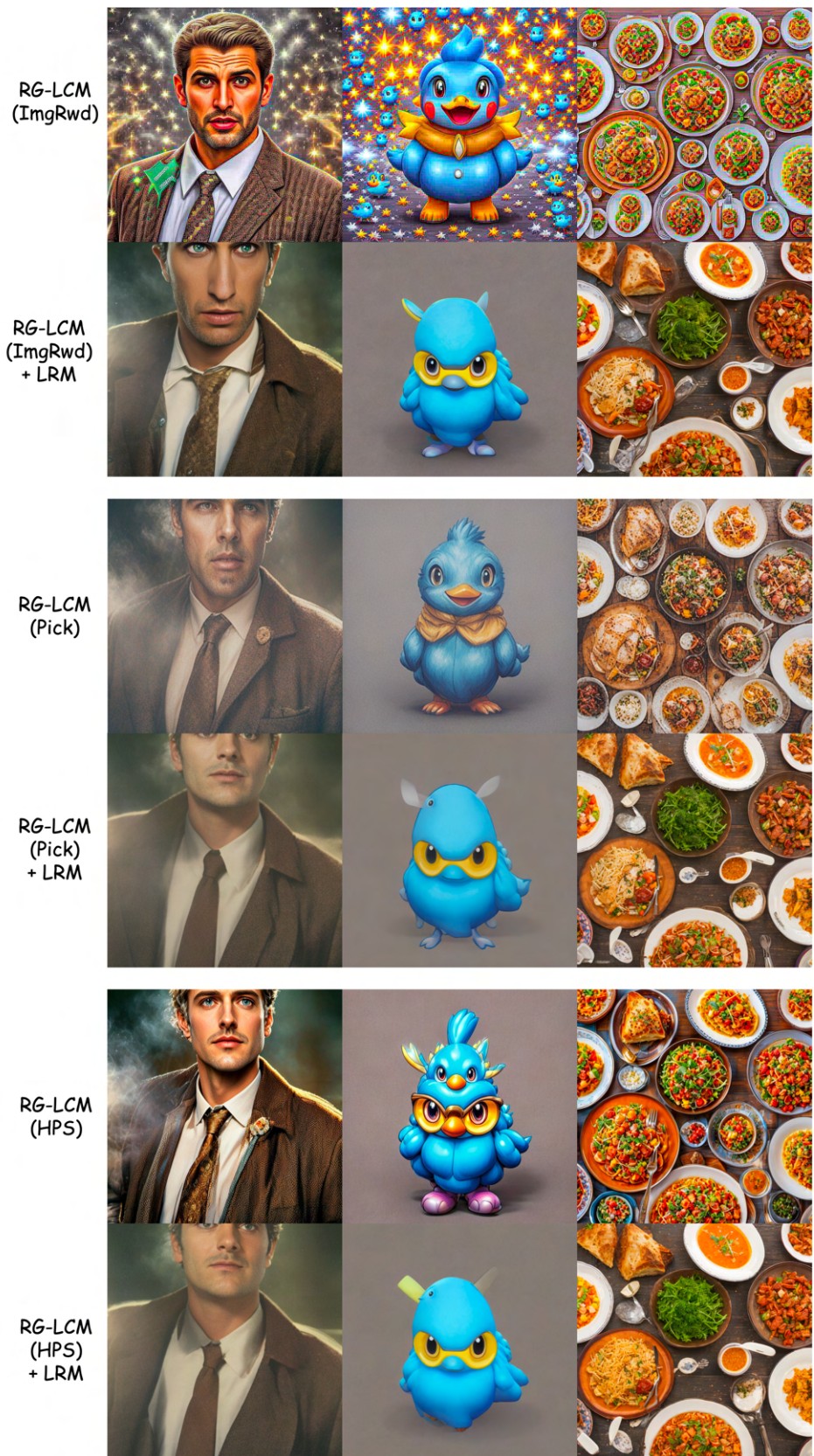

Figure 11: Effect of the Latent proxy RM (LRM). Integrating the LRM into our RG-LCD procedures makes the generated images natural, corresponding to the lower FID in Table 1. Moreover, it helps eliminate the high-frequency noise in the generated images.

RG-LCM (HPS) + LRM          LCM

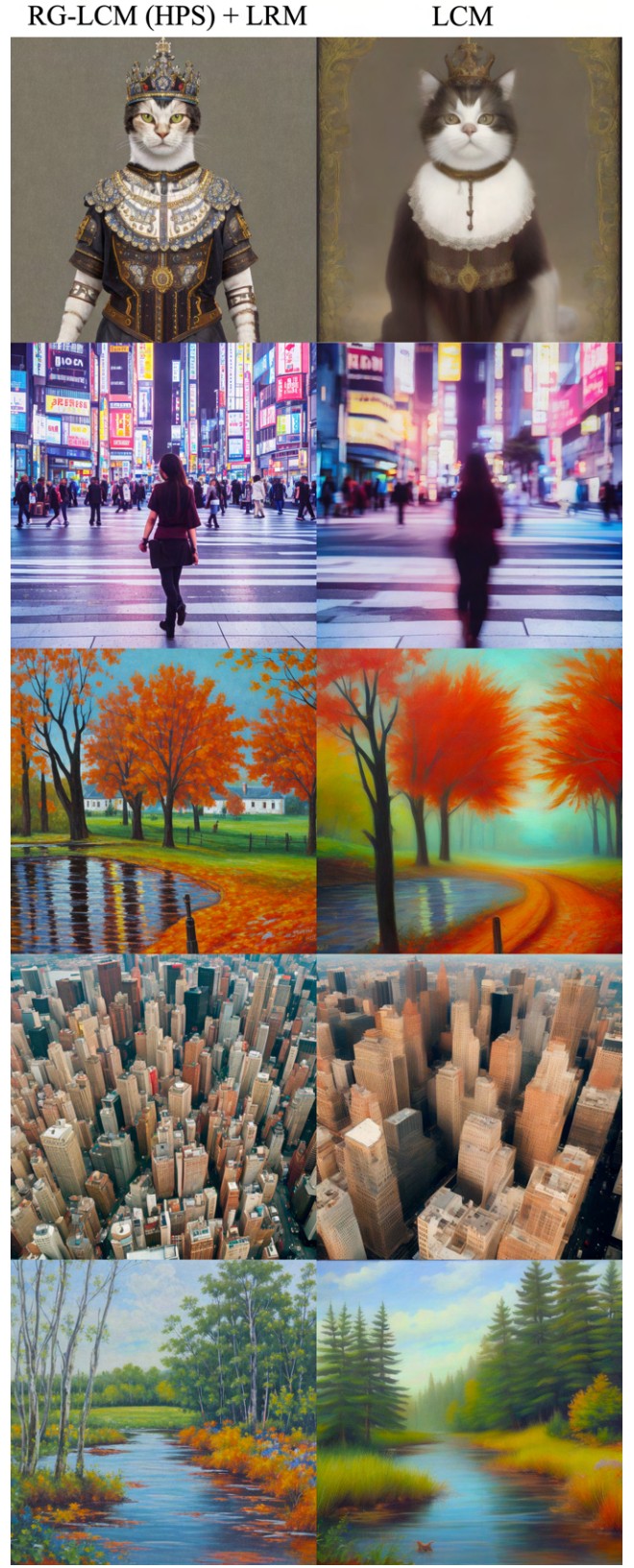

Figure 12: Comparison between RG-LCM (HPS) + LRM with LCM. The samples from RG-LCM (HPS) + LRM are visually appealing while remaining natural, corresponding to the high HPSv2.1 score and low FID in Table 1.

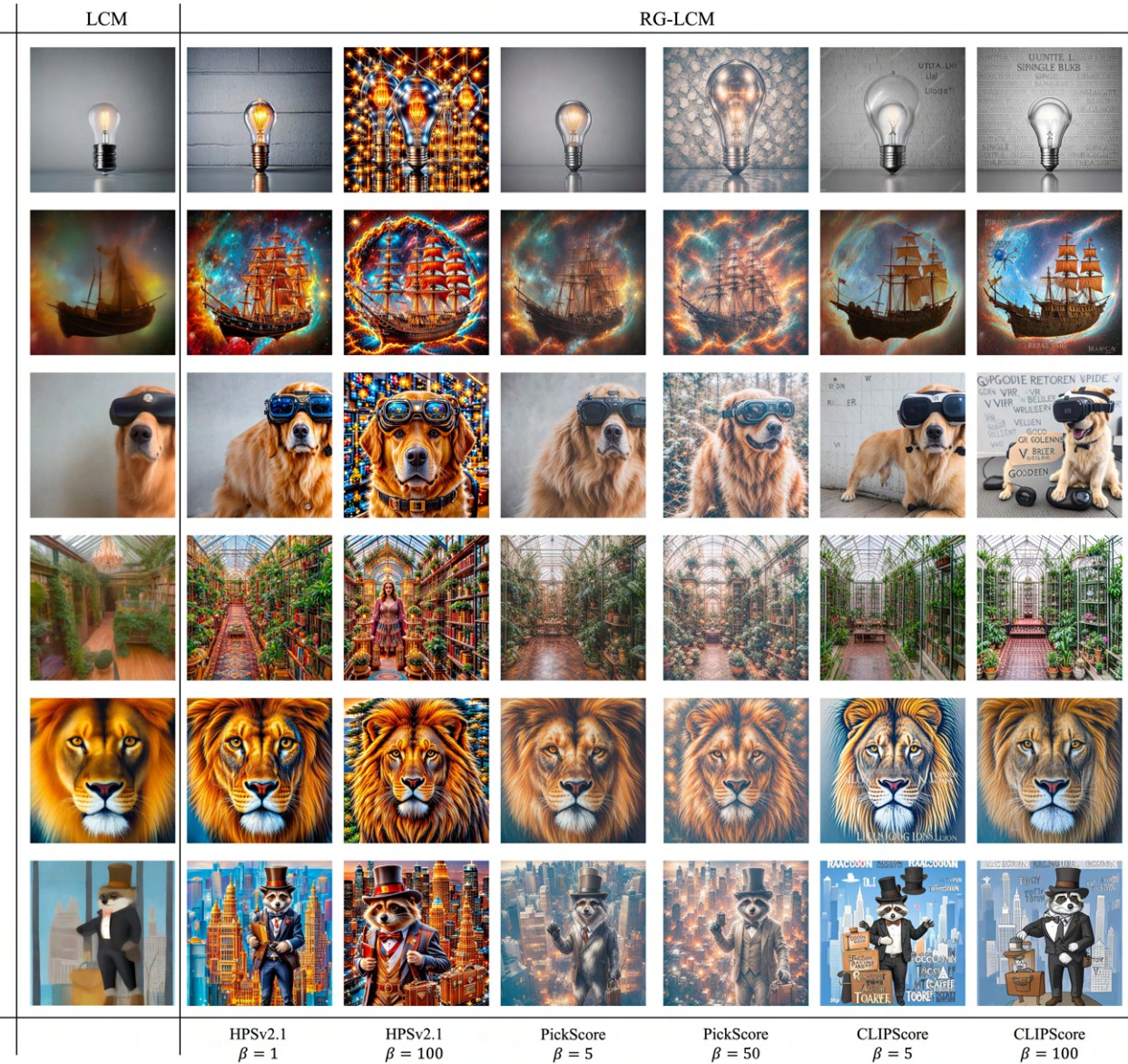

Figure 13: Additional images to study the impact of the reward scale $\beta$. We generate all samples with 4 steps.

Fig. 13 includes additional samples for the ablation on the reward scale $\beta$. The prompts in the top-to-bottom order are given below

- Ultra realistic photo of a single light bulb, dramatic lighting

- Pirate ship trapped in a cosmic maelstrom nebula

- A golden retriever wearing VR goggles.

- Highly detailed portrait of a woman with long hairs, stephen bliss, unreal engine, fantasy art by greg rutkowski.

- A stunning beautiful oil painting of a lion, cinematic lighting, golden hour light.

- A raccoon wearing a tophat and suit, holding a briefcase, standing in front of a city skyline.

Fig. 14 and 15 provide additional qualitative comparisons for RG-LCM (CLIP) with $\beta \in \{5, 100\}$, RG-LCM (HPS) with $\beta = 1$, and RG-LCM (HPS) + LRM with $\beta = 100$.

The prompts for Fig. 14 in the top-to-bottom order are given below

- A game screenshot featuring Woolie Madden with dreadlocks in Mass Effect.

- Two girls holding hands while watching the world burn

- A full-body portrait of a female cybered shadowrunner with a dark and cyberpunk atmosphere created by Echo Chernik in the style of Shadowrun Returns PC game.

- A portrait of a skeleton possessed by a spirit with green smoke exiting its empty eyes.

- A counter in a coffee house with choices of coffee and syrup flavors.

The prompts for Fig. 15 in the top-to-bottom order are given below

- 'Black and white portrait of Thabo Mbeki with highly detailed ink lines and a cyberpunk flair, created for the Inktober challenge as part of the Cyberpunk 2020 manual coloring pages.

- The image features a big white cliff, a cargo favela, a wall fortress, a neon pub, and some plants, with vivid and colorful style depicted in hyperrealistic CGI.

- An albino lion wearing a Mafia hat, digitally painted by multiple artists, trending on Artstation.

- A galaxy-colored DnD dice is shown against a sunset over a sea, in artwork by Greg Rutkowski and Thomas Kinkade that is trending on Artstation.

- A pirate skeleton.

Fig. 16 additional quantitative results for RG-LCM (ImgRwd) with images resized to a low resolution of 224x224. Please use Adobe Acrobat Reader and set the zoom to 100% (actual size). At this resolution, high-frequency noise becomes less noticeable. The prompts, listed in order from top to bottom and left to right, are provided below:

- A creepy cartoon rabbit wearing pants and a shirt, with dramatic lightning and a cinematic atmosphere.

- A beaver in formal attire stands beside books in a library.

- A pencil sketch of Victoria Justice drawn in the Disney style by Milt Kahl.

- Portrait of a male furry Black Reindeer anthro wearing black and rainbow galaxy clothes, with wings and tail, in an outerspace city at night while it rains.

- A galaxy-colored DnD dice is shown against a sunset over a sea, in artwork by Greg Rutkowski and Thomas Kinkade that is trending on Artstation.

- Architecture render with pleasing aesthetics.

- An empty road with buildings on each side.

- A computer monitor glows on a wooden desk that has a black computer chair near it.

- A man standing by his motorcycle is looking out to take in the view.

- A koala bear dressed as a ninja in a kayak.

- Baby Yoda depicted in the style of Assassination Classroom anime.

- A puppy is driving a car in a film still.

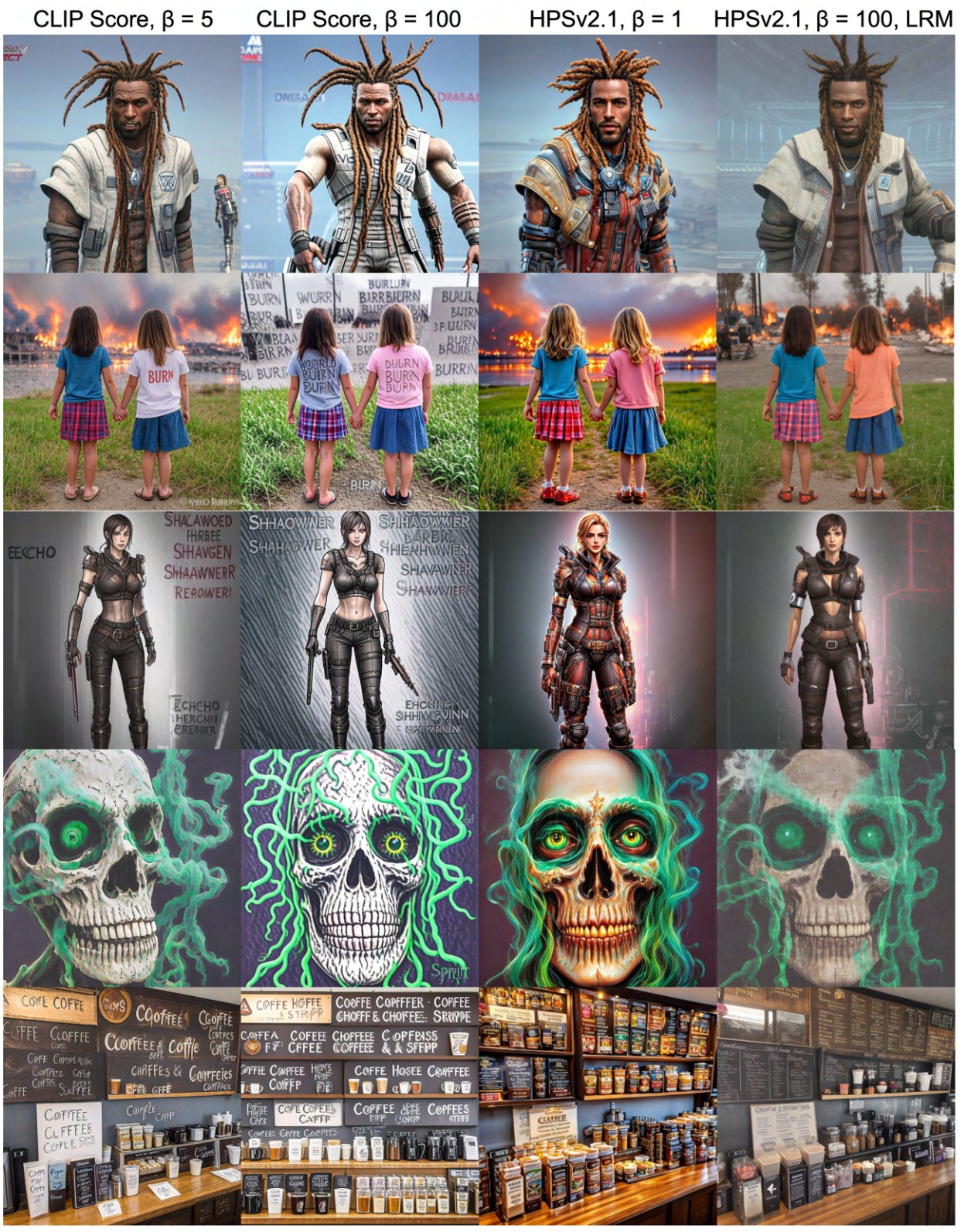

Figure 14: We present additional qualitative comparisons between RG-LCM (CLIP) with $\beta \in \{5, 100\}$, RG-LCM (HPS) with $\beta = 1$, and RG-LCM (HPS) + LRM with $\beta = 100$.

CLIP Score, β = 5    CLIP Score, β = 100    HPSv2.1, β = 1    HPSv2.1, β = 100, LRM

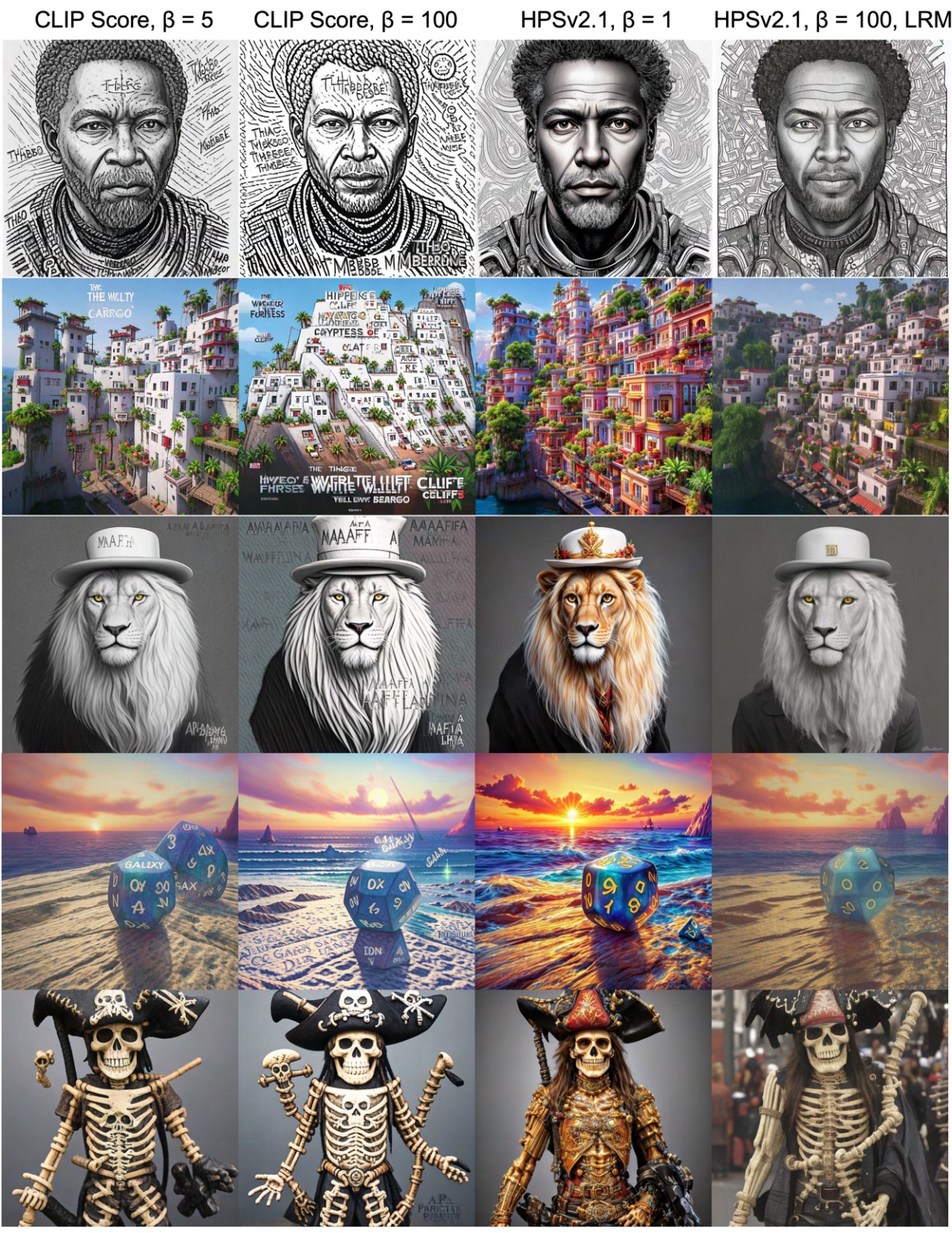

Figure 15: We present additional qualitative comparisons between RG-LCM (CLIP) with $\beta \in \{5, 100\}$, RG-LCM (HPS) with $\beta = 1$, and RG-LCM (HPS) + LRM with $\beta = 100$.

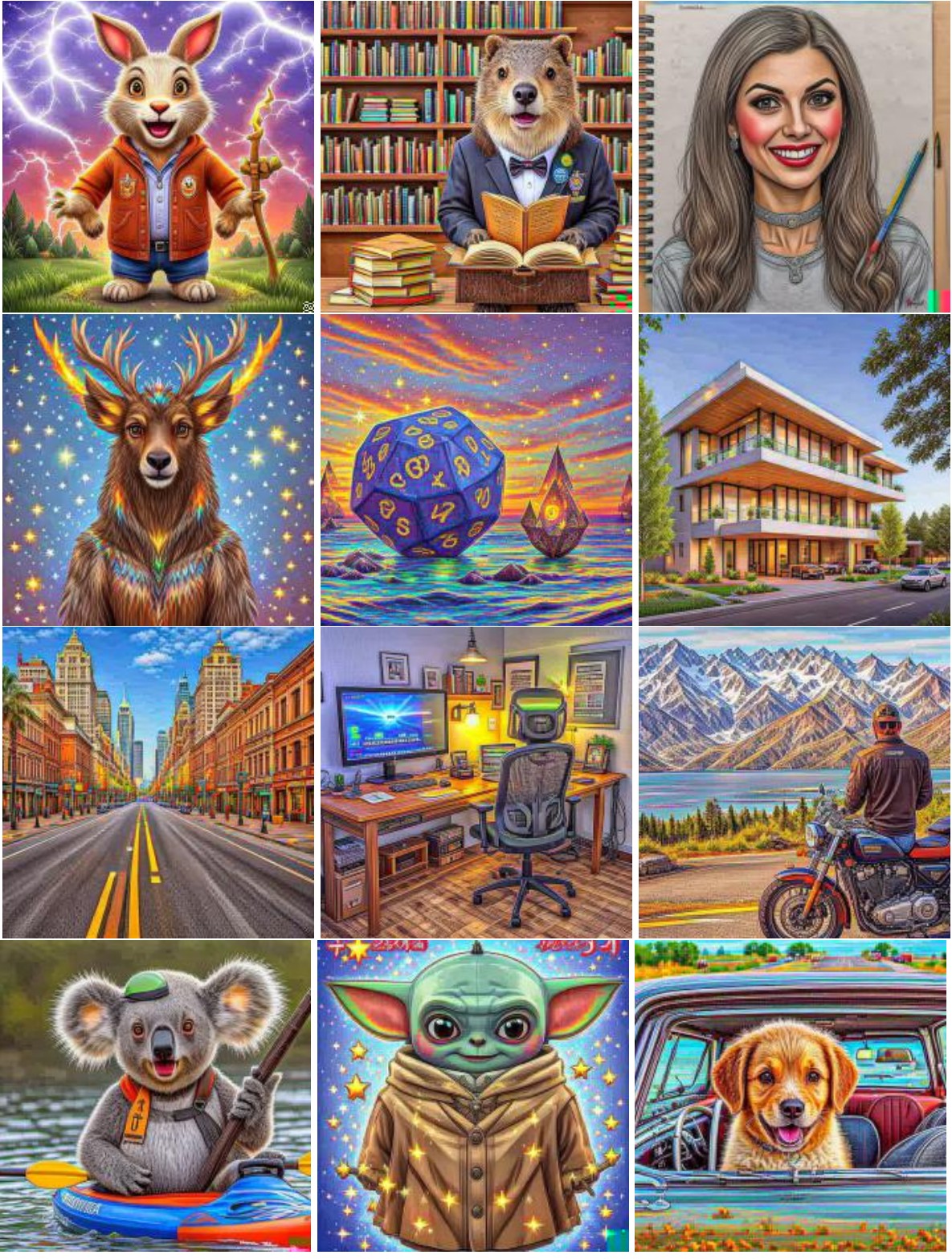

Figure 16: Additional quantitative results for RG-LCM (ImgRwd) with images resized to a low resolution of 224x224. Please use Adobe Acrobat Reader and set the zoom to 100% (actual size). At this resolution, high-frequency noise becomes less noticeable.

# D  Experiments with Additional Teacher T2I Models

In this section, we conduct additional experiments using different teacher T2I models, including Stable Diffusion 1.5 (SD 1.5) and Stable Diffusion XL (SDXL). For each teacher model, we train both the baseline LCM and our RG-LCM (HPS) by learning from HPSv2.1, using DDIM as defined in equation 13 as our ODE solver $\Psi$. The CC12M (Changpinyo et al., 2021) dataset serves as our training dataset. For SDXL, due to GPU memory constraints, we apply LCM-LoRA (Luo et al., 2023b) on top of SDXL to construct our RG-LCM and baseline LCM instead of performing full-model training. Additionally, we fix the weighting parameter $\beta$ in equation 11 to 1, consistent with the settings for Stable Diffusion 2.1.

Empirically, we evaluate different methods using 3,200 HPSv2 test prompts and employ VIEScore (Ku et al., 2023) as our evaluation metric with the GPT4o backbone. VIEScore achieves a high Spearman correlation of 0.4 with human evaluations, close to the human-to-human correlation of 0.45. Given a text-image pair, VIEScore provides *Semantic Score*, *Quality Score*, and *Overall Score*, reflecting text-image alignment, visual quality, and overall human preference, respectively. We compare the 4-step generation of our RG-LCM with the 4-step generation of the baseline LCM, as well as with the 4-step and 25-step generations from the teacher T2I model using DPM-Solver++ (Lu et al., 2022b) with CFG guidance (Ho & Salimans, 2022) and negative prompts. DPM-Solver++ is a high-order fast ODE solver that accelerates inference from diffusion models. It is important to note that we can also integrate DPM-Solver++ and negative prompts into our RG-LCM training. We leave it for future work.

Table 3 and 4 present the evaluation results. In both cases, the 4-step generation of our RG-LCM (HPS) outperforms other 4-step baselines. When using SD 1.5 as the teacher model, our 4-step generation even surpasses the 25-step generation achieved using DPM-Solver++ with CFG and negative prompts. With SDXL as the teacher model, our 4-step generation slightly underperforms but still matches the 25-step generation from the teacher. We believe this performance drop may be due to 1) the LoRA training and 2) the absence of high-quality image datasets. Therefore, we expect our RG-LCM to perform even better with full-model training and access to datasets with high image aesthetics, e.g., LAION-Aesthetics V2 6.5+ (Schuhmann et al., 2022).

| SD-1.5 as the Teacher Model | NFEs | Semantic Score ↑ | Quality Score ↑ | Overall Score ↑ |
|---|---|---|---|---|
| DPM-Solver++, Negative Prompt | 4 | 6.06 | 4.98 | 5.23 |
| DPM-Solver++, Negative Prompt | 25 | 6.77 | 6.63 | 6.45 |
| LCM | 4 | 6.75 | 5.88 | 6.02 |
| RG-LCM (HPS) | 4 | **7.55** | **7.02** | **7.11** |

Table 3: Evaluation of different methods using Stable Diffusion 1.5 as the teacher model on the HPSv2 test prompts. NFEs denote the number of function evaluations during inference. We employ VIEScore as the evaluation metric. By learning from the feedback of HPSv2.1, the 4-step generation of our RG-LCM (HPS) not only outperforms other 4-step baselines but also surpasses the 25-step generation achieved using DPM-Solver++ when sampling from the teacher model with CFG and negative prompts.

| SDXL as the Teacher Model | NFEs | Semantic Score ↑ | Quality Score ↑ | Overall Score ↑ |
|---|---|---|---|---|
| DPM-Solver++, Negative Prompt | 4 | 6.63 | 4.99 | 5.52 |
| DPM-Solver++, Negative Prompt | 25 | **8.23** | **7.73** | **7.83** |
| LCM | 4 | 7.26 | 6.43 | 6.65 |
| RG-LCM (HPS) | 4 | 8.1 | 7.46 | 7.64 |

Table 4: Evaluation of different methods using Stable Diffusion XL as the teacher model on the HPSv2 test prompts. NFEs denote the number of function evaluations during inference. We employ VIEScore as the evaluation metric. By learning from the feedback of HPSv2.1, the 4-step generation of our RG-LCM (HPS) not only outperforms other 4-step baselines but also matches the performance of the 25-step generation when using DPM-Solver++ to sample from the teacher model with CFG and negative prompts.

