# OpenReview forum: "Reward Guided Latent Consistency Distillation"
_TMLR — Accepted by TMLR_

### Review · Reviewer_UYSc · 2024-08-17

**Summary Of Contributions:**

This work addresses the quality loss by aligning LCM’s output with the human preference reward model during training. A reward-guided latent consistency distillation framework is proposed to enhance the LCM with reward model feedback. The core contribution addresses the reward overestimation issue due to direct optimization with the gradient from the RM. With the proposed method, 2-step diffusion inference can achieve better results than the classical 50 steps on SDv2.1.

**Audience:**

Yes

**Broader Impact Concerns:**

The impact statement has been discussed.

**Claims And Evidence:**

Yes

**Requested Changes:**

Comparisons on some other base models (SDXL, SD1.5), samplers (such as DPM-like samplers), with some wildly used techniques (such as CFG with certain negative prompts).

More visual results on the comparisons between RG-LCM (CLIP), RG-LCM (CLIP) + LRM, RG-LCM (HPS), and RG-LCM (HPS) + LRM.

A better evaluation metric would be more convenient.

**Strengths And Weaknesses:**

Pons:

- The idea of involving reward in the LCM distillation process is interesting. The core contribution is to address the reward overestimation issue by the use of a latent proxy RM (LRM). The technical novelty is moderate.

- The proposed strategy can work on the adopted models and achieve superior performance compared to the original LCM.

- User studies are performed on PartiPrompt and HPSv2 test sets for each baseline comparison. The ablation

Cons:

My main concerns are raised from the evaluations.

- The baselines are the SDv2.1 plus DDIM and LCM. While the best performance is usually achieved based on some other base models (SDXL, SD1.5), samplers (such as DPM-like samplers), and some other wildly used techniques (such as CFG with certain negative prompts). I wonder if the proposed method can achieve comparable performance on the commonly used settings, or it only works on the standard  SDv2.1 + DDIM/LCM settings which are quite limited for the community.

- It is mentioned in Table 1 that “Except trained with CLIPScore, our RG-LCMs achieve better”, where I suppose the CLIP reward model can achieve good results without the proposed RG-LCMs. Besides, as illustrated in Figure 6, I think the CLIPScore with Beta=100 achieves the best quality. HPSv2.1 leads to color over-saturation issues, and PickScore leads to haze issues. More visual results on the comparisons between RG-LCM (CLIP), RG-LCM (CLIP) + LRM, RG-LCM (HPS), and RG-LCM (HPS) + LRM would be helpful.

- It is also questionable if the HPSv2.1 metric can reflect the quality of different method results. Besides, the HPSv2.1 score is used to boost model performance in the RG-LCM (HPS) + LRM. It could be a bit unfair in this evaluation.
Human evaluation is a good solution. I would also recommend using VLM models for evaluation (like GPT4V or miniGPT4 adopted in [1])

- It is unclear what the D means in Section 4.1. The explanation should be provided following equation (10).

[1] T2I-CompBench++: An Enhanced and Comprehensive Benchmark for Compositional Text-to-image Generation

Minor:

I suppose there is a typo in section 3.1: where µ(·) and µ(·) are the drift and diffusion

---

> ### Author Response · Authors · 2024-09-01
> **Response to Reviewer UYSc**
>
> We thank the constructive feedback from the reviewer!
>
> > I wonder if the proposed method can achieve comparable performance on the commonly used settings, or it only works on the standard SDv2.1 + DDIM/LCM settings.
>
> Following the request of the reviewer,  we conduct additional experiments using SD 1.5 and SDXL as the teacher T2I models in Appendix D.  To demonstrate the effectiveness of our method,  we compare the 4-step generation of our RG-LCM with the 4-step generation of the baseline LCM, as well as with the 4-step and 25-step generations from the teacher T2I model using DPM-Solver++ with CFG guidance and negative prompts. We use the 3,200 HPSv2 test prompts and VIEScore (GPT4o) [1] as the metric.
>
> As shown in Table 3 and 4, in both cases, the 4-step generation of our RG-LCM (HPS) outperforms other 4-step baselines. When using SD 1.5 as the teacher model, our 4-step generation even surpasses the 25-step generation achieved using DPM-Solver++ with CFG and negative prompts. With SDXL as the teacher model, our 4-step generation slightly underperforms but still matches the 25-step generation from the teacher. We believe this performance drop may be due to 1) the LoRA training and 2) the absence of high-quality image datasets. Therefore, we expect our RG-LCM to perform even better with full-model training and access to datasets with high image aesthetics, e.g., LAION-Aesthetics V2 6.5+.
>
> [1] Ku et al. VIEScore: Towards Explainable Metrics for Conditional Image Synthesis Evaluation. ACL 2024
>
> > It is mentioned in Table 1 that “Except trained with CLIPScore, our RG-LCMs achieve better”, where I suppose the CLIP reward model can achieve good results without the proposed RG-LCMs.
>
> We would like to clarify that when trained with CLIPScore, our RG-LCM (CLIP) achieves a higher HPSv2.1 Score than the baseline LCM without increasing the FID value on the MS-COCO dataset. This indicates that our RG-LCM (CLIP) is capable of generating high-quality images that are not only preferred by humans but also closely aligned with the real images in the MS COCO dataset.
>
> It's important to note that CLIPScore functions as a reward model, whereas our RG-LCM is a T2I model. We kindly ask the reviewer to clarify their question so we can address it more accurately.
>
> > Besides, as illustrated in Figure 6, I think the CLIPScore with Beta=100 achieves the best quality. HPSv2.1 leads to color over-saturation issues, and PickScore leads to haze issues. More visual results on the comparisons between RG-LCM (CLIP), RG-LCM (CLIP) + LRM, RG-LCM (HPS), and RG-LCM (HPS) + LRM would be helpful.
>
> First, we would like to clarify that we did not report results for RG-LCM (CLIP) + LRM in Table 1 because RG-LCM (CLIP) already achieves a higher HPSv2.1 Score without increasing the FID value compared to the baseline LCM.
>
> In Fig. 14 and 15 of Appendix C, we provide additional qualitative comparisons between RG-LCM (CLIP) with $\beta\in\{5, 100\}$, RG-LCM (HPS) with $\beta = 1$, and RG-LCM (HPS) + LRM with $\beta = 100$. As shown in the figures, setting $\beta = 100$ for RG-LCM (CLIP) results in images where the text prompts are visibly incorporated into the imagery. It can also lead to repeated patterns. Both phenomena align with our findings in Fig. 6. Additionally, while HPSv2.1 can sometimes lead to color over-saturation, incorporating our LRM helps to alleviate this issue.
>
> > I would also recommend using VLM models for evaluation (like GPT4V or miniGPT4)
>
> For our newly included experiments in Appendix D, we report their VIEScore on the 3,200 HPSv2 test prompts. With the GPT4o backbone, VIEScore achieves a high Spearman correlation of 0.4 with human evaluations, close to the human-to-human correlation of 0.45. Given a text-image pair, VIEScore provides `Semantic Score`, `Quality Score`, and `Overall Score`, reflecting text-image alignment, visual quality, and overall human preference, respectively. As shown in our new results in Tables 3 and 4, the 4-step samples from our RG-LCM (HPS) outperform the other 4-step baselines across all three VIEScore dimensions and at least match, if not outperform, the 25-step generation from the teacher model when using SD 1.5 and SDXL as the teacher T2I models.
>
> > It is unclear what the D means in Section 4.1. The explanation should be provided for eq. 10
>
> We thank the reviewer for pointing this out. The $\mathcal{D}$ in Section 4.1 denotes the VAE decoder.
>
> > Comparisons on some other base models (SDXL, SD1.5), samplers (such as DPM-like samplers), with some wildly used techniques (such as CFG with certain negative prompts).
>
> We have included the experiments in Appendix D.
>
> > More visual results on the comparisons between RG-LCM (CLIP), RG-LCM (CLIP) + LRM, RG-LCM (HPS), and RG-LCM (HPS) + LRM.
>
> We have included the requested qualitative comparisons in Fig. 14 and 15 of Appendix C.
>
> > A better evaluation metric would be more convenient.
>
> We employ VIEScore (GPT4o) as our evaluation metric for our new experiments.

---

### Review · Reviewer_a19z · 2024-08-18

**Summary Of Contributions:**

This paper introduces Reward Guided Latent Consistency Distillation (RG-LCD), a method that integrates feedback from a reward model (RM) into the Latent Consistency Distillation (LCD) process for text-to-image synthesis. The authors propose compensating for the quality loss of LCD by aligning the output of the latent consistency model (LCM) with human preference during training. They demonstrate that when trained with the feedback of a good RM, the RG-LCM achieves high-quality image generation with a significant inference acceleration. They also propose the use of a latent proxy RM (LRM) to overcome reward over-optimization. Experimental results show that incorporating the LRM into RG-LCD improves image quality and outperforms the baseline LCM.

**Audience:**

Yes

**Broader Impact Concerns:**

The broader impact of this work is generally positive, as it enables faster and more efficient generation of high-quality images, which can benefit various applications, from digital art to content creation. However, there are some concerns regarding the potential misuse of such technology in generating misleading or harmful content, especially if the reward models are not properly aligned with ethical considerations. We hope author can provide more insightful discussion in this part.

**Claims And Evidence:**

Yes

**Requested Changes:**

1. We hope the author can provide detailed training process of latent proxy RM. Including the pseudo code would help in comprehending how the LRM is trained.
2. Provide more discussion on how to choose the reward function and its influence.

**Strengths And Weaknesses:**

Strengths:
1. The paper introduces a novel method, RG-LCD, which addresses the quality loss issue in LCD and achieves high-quality image generation with significant inference acceleration.
2. The use of a reward model and the integration of feedback from the RM into the LCD process are innovative and practical approaches to improve the sample quality. The introduction of a latent proxy RM (LRM) as an intermediary to connect the LCM with the RM is a clever solution to avoid reward over-optimization.
3. The experimental results demonstrate the effectiveness of the proposed method, with improved image quality and outperformance of the baseline LCM.

Weakness:
1. Clarify the Training Process of Latent Proxy RM: The paper does not clearly explain the process of training the Latent Proxy RM, making it difficult to understand the exact steps involved. A more detailed and structured explanation or pseudo code would help in comprehending how the LRM is trained and integrated into the RG-LCD framework.
2. The reward function appears to be heuristic, with many available options that significantly impact the results. We hope author provide more discussion on this part.

---

> ### Author Response · Authors · 2024-09-01
> **Response to Reviewer a19z**
>
> We thank the reviewer for the constructive feedback! Please find our detailed responses below.
>
> > Clarify the Training Process of Latent Proxy RM
>
> The training of our Latent Proxy RM (LRM) contains two stages: 1) The pretraining and 2) the finetuning stage.
>
> In this initial pretraining stage, the goal is to pretrain the image latent encoder to align it with the pretrained CLIP text encoder. This process mirrors the CLIP pretraining pipeline and utilizes contrastive loss as the training objective.
>
> During the finetuning stage, we align the LRM $\mathcal{R}\_{\sigma}^L$ with the preferences of an expert RM $\mathcal{R}^E$. This alignment is integrated into our RG-LCD framework. Note that computing the $L\_\text{LCD}$ (Eq. 8) requires computing $\mathbf{z}\_1 = \boldsymbol{f}\_\theta(\mathbf{z}\_{t_{n+k}}, \omega, \mathbf{c}, t_{n+k})$ and $\mathbf{z}\_2 = \boldsymbol{f}\_\theta(\mathbf{z}\_{t_{n}}, \omega, \mathbf{c}, t_{n})$. Alongside the latent $\mathbf{z}_0$ derived from the training image $\mathbf{I}_0$, these components serve as the essential inputs for training our $\mathcal{R}\_{\sigma}^L$. It is important to note that all three images share the same text description $\mathbf{c}$. Note that **we have updated Eq. 12 in our manuscript to improve its clarity**.
>
> We first obtain $\mathcal{R}^E$'s rewards on these latents as $\mathcal{R}^E(\mathcal{D}(\mathbf{z}\_0), \mathbf{c})$, $\mathcal{R}^E(\mathcal{D}(\mathbf{z}\_1), \mathbf{c})$ and $\mathcal{R}^E(\mathcal{D}(\mathbf{z}\_2), \mathbf{c})$, where $\mathcal{D}$ is the VAE decoder. Similarly, $\mathcal{R}\_{\sigma}^L$'s rewards on these latents can be by $\mathcal{R}\_{\sigma}^L(\mathbf{z}\_0, \mathbf{c})$, $\mathcal{R}\_{\sigma}^L(\mathbf{z}\_1, \mathbf{c})$ and $\mathcal{R}\_{\sigma}^L(\mathbf{z}\_2, \mathbf{c})$. Finally, we minimize the learning loss $L_{RM}(\sigma)$ (Eq. 12), which consists of three KL divergence terms, to align $\mathcal{R}_{\sigma}^L$'s preferences with those of the expert RM $\mathcal{R}^E$.
>
> Additionally, **we have updated our manuscript to include detailed pseudo-codes for our RG-LCD with a LRM in Algorithm 4, Appendix B.3**.
>
> > The reward function appears to be heuristic, with many available options that significantly impact the results. We hope author provide more discussion on this part.
>
> We assume the reviewers are using the "reward function" and "reward model (RM)" interchangeably. Therefore, **we interpret the question as inquiring about how to choose the most suitable reward model for training our RG-LCM**. If this assumption is incorrect, please feel free to clarify, and we will adjust our discussion accordingly.
>
> There are several approaches to selecting a reward function. As demonstrated in our paper, one option is to use off-the-shelf RMs such as HPSv2.1, PickScore, and ImageReward to train the RG-LCM. Additionally, a linear combination of these different RMs, as explored in [1], could be employed.
>
> Alternatively, any vision-language model can serve as the expert RM $\mathcal{R}^E$, with the RG-LCM trained using our LRM as a proxy. We believe this latter approach offers a promising future direction, as it enables efficient learning from any RGB-based models without requiring their differentiability. For instance, $\mathcal{R}^E$ could be set as a multimodal large language model (MLLM), such as GPT-4 or Gemini, or other non-differentiable RMs, such as LLMScore, VIEScore, and DA-Score. These models are generally pretrained on a wide array of vision-language corpora and contain extensive knowledge. In contrast, off-the-shelf RMs like HPSv2.1, PickScore, and ImageReward are limited by their specific training data. Therefore, learning from these advanced models holds the potential to induce a more advanced T2I model, which we leave it to future work.
>
> [1] Clark et al. Directly Fine-Tuning Diffusion Models on Differentiable Rewards. ICLR 2024.
>
> > Potential misuse of such technology in generating misleading or harmful content
>
> While the primary goal of this research is to advance the generation of high-quality images for positive applications such as digital art, content creation, and other creative industries, we acknowledge the potential risks associated with the misuse of this technology.
>
> One key concern is generating misleading or harmful content, which could arise if the reward models are not carefully aligned with ethical guidelines. To mitigate this risk, we emphasize the importance of developing and incorporating ethical frameworks into designing and deploying reward models. This includes ensuring that the models are trained on diverse, representative, and ethically sourced datasets and regularly audited for biases or tendencies that could lead to harmful outcomes.
>
> Furthermore, we will perform user controls when releasing our training codes and models to prevent the misuse of this technology. We will implement ethical guidelines for content generation to ensure the technology is used responsibly.

---

### Review · Reviewer_ZfZw · 2024-08-21

**Summary Of Contributions:**

This paper introduces fine-tuning the few-step generator with auxiliary loss that is computed at the latent space. The proposed auxiliary loss, the negative of reward surrogate function, deviates the denoising trajectory from the trajectory from PF-ODE, in preferable directions.

**Audience:**

Yes

**Broader Impact Concerns:**

The deep fake generation is always on the concerning list in this kind of works. However, it would be unfair to impose such ethical concern for this particular paper. I would say there is no ethical concern for this paper.

**Claims And Evidence:**

No

**Requested Changes:**

1. ImageGen -> Imagen
2. Eq. (11) should be re-written for the reward surrogate function at the latent space. It is extremely confusing to understand.
3. I don't understand Eq. (12). What does i and j denote for?
4. Replace $\sigma$ into $\phi$ or something else that most people get immediately as the parameter. $\sigma$ is usually designated for the noise level in diffusion community.
5. What does $E$ stand in $R^{E}$? Please recheck all terms carefully and define properly.

**Strengths And Weaknesses:**

Strength
- It is good to see that the suggested algorithm is very clear while performing good quality.

Weakness
- It is unclear why latent reward function can resolve the high-frequency issue presented in Figure 3. Simply saying replacing the evaluation from pixel space to latent space solves the issue does not make sense to me. I need more experimentally grounded or theoretically solid evidence for this. This is not the novelty concern. This is rather the correctness and completeness concern.
- Let's suppose that the CLIP text encoder is infinitely flexible and we have amazing optimizer. If my understanding is correct, if $\tau_{L}$ and $\tau_{E}$ are the same, then the optimal solution of $L_{RM}(\sigma)$ (when we optimize Eq. (12)) should be $R_{\sigma}^{L}(z,c)=R^{E}(D(z),c)$. In that optimal solution, Figure 3 will happen again. Correct me if I'm wrong.
- Do you have any experimental result if you fine-tune too much the CLIP text encoder? If it falls into the global optimum, $L_{RM}(\sigma)=R_{\sigma}^{L}(z,c)=R^{E}(D(z),c)$, and Figure 3 will happen again. What is happening in the optimization procedure? Either algorithmic explanation would also be possible if it is plausible.

---

> ### Author Response · Authors · 2024-09-01
> **(1/2) Response to Reviewer ZfZw**
>
> > It is unclear why latent reward function can resolve the high-frequency issue presented in Figure 3. Simply saying replacing the evaluation from pixel space to latent space solves the issue does not make sense to me. I need more experimentally grounded or theoretically solid evidence for this. This is not the novelty concern. This is rather the correctness and completeness concern.
>
> We would like to clarify a misunderstanding in the review. We did NOT claim that replacing the evaluation from pixel space to latent space solves the issue of high-frequency noise. Instead, we identified that the _Resize_ operation during the preprocessing phase causes the reward model (RM) to overlook high-frequency noise. In other words, if a well-trained RM does not require resizing the input image to a lower resolution, it is more likely to detect and penalize high-frequency noise in the corresponding images. As shown in Fig.3, the high-frequency noise becomes less perceptible when images are reduced in size.
>
> > Let's suppose that the CLIP text encoder is infinitely flexible and we have amazing optimizer. If my understanding is correct, if  $\tau_L$  and  $\tau_E$  are the same, then the optimal solution of  $L_{RM}(\sigma)$  (when we optimize Eq. (12)) should be  $\mathcal{R}_\sigma^L(z,c)=\mathcal{R}^E(\mathcal{D}(z),c)$. In that optimal solution, Figure 3 will happen again. Correct me if I'm wrong.
>
> We would like to emphasize that a key advantage of our versatile LRM is its ability to **mitigate reward overoptimization** towards the expert $\mathcal{R}^E$. Our goal is _NOT_ to achieve the optimal solution of $L_{RM}(\sigma)$ by training $\mathcal{R}\_\sigma^L$ to exactly match the imperfect $\mathcal{R}^E$. If we were to do so, $\mathcal{R}\_\sigma^L$ would also learn to ignore the high-frequency noise present in the input image. Given a differentiable $\mathcal{R}^E$, using our LRM would be redundant and inefficient if the objective were to fully optimize $\mathcal{R}^E$, as directly leveraging $\mathcal{R}^E$'s gradients would be more efficient in that scenario.
>
> In summary, our LRM $\mathcal{R}\_\sigma^L$ helps to mitigate the risk of overestimating rewards from an imperfect $\mathcal{R}^E$. Additionally, if $\mathcal{R}^E$ provides perfect rewards in an ideal scenario but is non-differentiable, $\mathcal{R}\_\sigma^L$ can serve as an intermediary, bridging the non-differentiable gap between our RG-LCM and $\mathcal{R}^E$. In such cases, we can employ various optimization strategies, such as a dedicated learning phase to align $\mathcal{R}\_\sigma^L$ with $\mathcal{R}^E$ before conducting our RG-LCD training, to help $\mathcal{R}\_\sigma^L$ better match the perfect $\mathcal{R}^E$.
>
> > Do you have any experimental result if you fine-tune too much the CLIP text encoder? If it falls into the global optimum,  $L_{RM}(\sigma) = R_\sigma^L(z,c)=\mathcal{R}^E(\mathcal{D}(z),c)$, and Figure 3 will happen again. What is happening in the optimization procedure? Either algorithmic explanation would also be possible if it is plausible.
>
> To address the concern about over-finetuning the CLIP text encoder, we conducted an experiment where we trained an RG-LCM (ImgRwd) + CLIP model without freezing any layers of the CLIP text encoder. In our previous experiments, we followed the HPSv2 approach and only finetuned the last five layers of the CLIP text encoder. Even with extensive finetuning (i.e., without freezing any layers), the resulting RG-LCM did not produce images with high-frequency noise.
>
> > ImageGen -> Imagen
>
> Thanks for pointing it out. We have fixed the typo.
>
> > Eq. (11) should be re-written for the reward surrogate function at the latent space. It is extremely confusing to understand.
>
> **Could you elaborate on the request?** We don't see why Eq. 11 is confusing in its current form. We assume you are requesting to add a similar objective for our RG-LCD + LRM. To improve its clarity, we have instead included pseudo-codes for RG-LCD + LRM training in Algorithm 4 of Appendix B.3.
>
> > Replace  $\sigma$  into  $\phi$ or something else that most people get immediately as the parameter.  $\sigma$  is usually designated for the noise level in diffusion community.
>
> We appreciate your feedback. However, we have used $\phi$ and $\psi$ to denote the parameters of the ODE solver in Eq. 7 and 9. We acknowledge that using $\sigma$ for LRM's parameters might cause some confusion. We plan to change the notation from $\sigma$ to $\zeta$ after this rebuttal period to prevent any confusion for other reviewers.
>
> > What does  E  stand in  $R^E$? Please recheck all terms carefully and define properly.
>
> The $E$ stands for "expert" in $\mathcal{R}^E$. We will ensure all terms are clearly defined in the final version of the manuscript.

---

> ### Author Response · Authors · 2024-09-01
> **(2/2) Response to Reviewer ZfZw**
>
> >  I don't understand Eq. (12). What does i and j denote for?
>
> Thanks for pointing it out. **We have revised Equation 12 to improve clarity and correct the notation**. When computing the $L_{RM}(\sigma)$, we use the latnets $\mathbf{z}\_0$, $\mathbf{z}\_1 = \boldsymbol{f}\_\theta(\mathbf{z}\_{t_{n+k}}, \omega, \mathbf{c}, t_{n+k})$ and $\mathbf{z}\_2 = \boldsymbol{f}\_\theta(\mathbf{z}\_{t_{n}}, \omega, \mathbf{c}, t_{n})$. Here, $i$ and $j$ are used to denote the index for these latents.
>
> Our formulation of $L_{RM}(\sigma)$ is inspired by the learning loss of HPSv2, and the underlying concept is intuitive. Given $\mathbf{z}\_0$, $\mathbf{z}\_1$ and $\mathbf{z}\_2$ corresponding to the same text description $\mathbf{c}$, we can group them into three pairs: $(\mathbf{z}\_0, \mathbf{z}\_1)$,  $(\mathbf{z}\_0, \mathbf{z}\_2)$  and $(\mathbf{z}\_1, \mathbf{z}\_2)$. We then use $\mathcal{R}\_\sigma^L$ and $\mathcal{R}^E$ to compute the rewards for each latent. For each latent pair $(\mathbf{z}\_i, \mathbf{z}\_j)$, the probability of $\mathcal{R}\_\sigma^L$ preferring $\mathbf{z}\_i$ over $\mathbf{z}\_j$ is modeled as:
>
> $$P^\sigma_{i,j}(i) = \frac{\exp\left(\mathcal{R}^{\text{L}}\_\sigma\left(\mathbf{z}\_i, \mathbf{c} \right) / {\tau_L}\right)}{\exp\left(\mathcal{R}^{\text{L}}\_\sigma\left(\mathbf{z}\_i, \mathbf{c} \right) / {\tau_L}\right) + \exp\left(\mathcal{R}^{\text{L}}\_\sigma\left(\mathbf{z}\_j, \mathbf{c} \right) / {\tau_L}\right)}$$
>
> Similarly, the probability of $\mathcal{R}^E$ preferring $\mathbf{z}\_i$ over $\mathbf{z}\_j$ is modeled as:
>
> $$Q_{i,j}(i) = \frac{\exp\left(\mathcal{R}^E\left(\mathcal{D}\left(\mathbf{z}\_i\right), \mathbf{c} \right) / {\tau_E}\right)}{\exp\left(\mathcal{R}^E\left(\mathcal{D}\left(\mathbf{z}\_i\right), \mathbf{c} \right) / {\tau_E}\right) + \exp\left(\mathcal{R}^E\left(\mathcal{D}\left(\mathbf{z}\_j\right), \mathbf{c} \right) / {\tau_E}\right)}$$
>
> And thus, we have
> $$P^\sigma_{i,j}(m) \propto \exp\left(\mathcal{R}^{\text{L}}\_\sigma\left(\mathbf{z}\_m, \mathbf{c} \right) / {\tau_L}\right), Q_{i,j}(m) \propto \exp\left(\mathcal{R}^E\left(\mathcal{D}\left(\mathbf{z}\_m\right), \mathbf{c} \right) / {\tau_E}\right), m\in\{i, j\}$$
>
> Finally, we construct the KL divergence between the distribution $P^\sigma_{i,j}$ and $Q_{i,j}(i)$ for each $(\mathbf{z}\_i, \mathbf{z}\_j)$ pair as $D_{\text{KL}}\left(P^\sigma_{i,j} || Q_{i,j}\right)$. Our $L_{RM}(\sigma)$ is derived by summing the KL divergence for all three latent pairs.

---

### Author Response · Authors · 2024-09-01
**New Experiment Results!**

We appreciate the reviewers' time and constructive feedback on our work. We have responded to individual reviews below and updated the manuscript accordingly. Here, we would like to highlight some new results and changes that may be of interest:

1. We conducted additional experiments using SD 1.5 and SDXL as the teacher T2I models, as detailed in Appendix D, and performed automatic evaluation using the VIEScore (GPT4o) metric.

2. We revised Equation 12 to improve clarity and corrected some notation errors. Changes made throughout the paper are highlighted in blue font for easy reference.

---

### Comment · Action_Editor_ztnq · 2024-09-19
**Request**

Dear authors,

One of the reviewers has requested addressing the following questions (you may further update the PDF if necessary):

(1) By what sense does the switching from pixel-space optimization to latent-space optimization resolves the high-frequency issue?

(2) At which criteria should we stop the finetuning?  I [the reviewer] can get that the infinite iterations of finetuning may ruin the generation, but I think it's then crucial to suggest the stopping criteria.

Regards,

Action Editor

---

> ### Author Response · Authors · 2024-09-22
> **Response to the follow-up questions**
>
> > (1) By what sense does the switching from pixel-space optimization to latent-space optimization resolves the high-frequency issue?
>
> We would like to clarify that utilizing a latent reward model (RM) that operates in latent space eliminates the need for the **resize** operation during the preprocessing phase. A robust VAE encoder-decoder pair ensures that the latent embedding retains the essential information of its corresponding RGB image, as the VAE decoder can accurately reconstruct the image. This allows the latent reward RM to capture high-frequency noise in the input image latent effectively. In contrast, a conventional RGB-based RM, constrained by a fixed input size (e.g., 224 x 224), requires large images to be **resized**, leading to **a loss of high-frequency details**. As a result, the RGB-based reward model, operating on low-resolution inputs, may overlook the high-frequency noise in the original images. This is supported by both our qualitative results in Figure 3 and quantitative results in Table 1, where the images generated by RG-LCM (ImgRwd), despite containing high-frequency noise, still receive a high score from the RM HPSv2.1.
>
> > (2) At which criteria should we stop the finetuning? I [the reviewer] can get that the infinite iterations of finetuning may ruin the generation, but I think it's then crucial to suggest the stopping criteria.
>
> In practice, we use a large and diverse text-image dataset, such as CC12M, for RG-LCD training. In this paper, we fixed the training to 10K iterations and observed stable performance without encountering any training instability. We hypothesize that performance degradation would only occur if training exceeds one full epoch of the dataset. However, even with a large batch size of 256, one epoch would require 12M / 256 = 47.9K iterations, which is far beyond the 10K iterations we used. Therefore, stopping criteria are not a significant concern for our method.
>
> That said, we could still define a stopping criterion for the latent RM by monitoring the average rewards of training batches. We can stop the LRM training when the average rewards converge to a specific value.

---

> > ### Comment · Reviewer_ZfZw · 2024-09-23
> > **Thanks to the authors**
> >
> > Thank you for the author's comments.
> >
> > (1) Could you put additional 224x224-resized figure? If the claim is correct, then the 224x224-resized figure with RGB-based RM should be perceptually good, with no checkerboard effect like in Figure 3. If it is not visually appealing, there is something hidden factor that makes the latent-based RM excel the RGB-based RM.
> >
> > (2) That's what I'm asking. Could you explicitly mention your argument about finetuning in the manuscript? (probably at the \textbf{Settings} of Section 5 or at the Appendix)

---

> > > ### Author Response · Authors · 2024-09-23
> > > **Response to the comment**
> > >
> > > Dear Reviewer ZfZw,
> > >
> > > We have updated the manuscript to address your further concerns. Please find our response below.
> > >
> > > > (1) Could you put additional 224x224-resized figure?
> > >
> > > In Fig. 16, we have provided additional quantitative results for RG-LCM (ImgRwd) with images resized to a low resolution of 224x224. Please use Adobe Acrobat Reader and set the zoom to 100\% (actual size). At this resolution, high-frequency noise and checkerboard effect become less noticeable.
> > >
> > > > Could you explicitly mention your argument about finetuning in the manuscript?
> > >
> > > We have included our argument about finetuning in Appendix A. The change is again highlighted in the blue font for easy reference.

---

> > > > ### Comment · Reviewer_ZfZw · 2024-09-23
> > > > **Thanks for the authors**
> > > >
> > > > I appreciate for the author's response.
> > > > My concerns are appropriately addressed.

---

> > > > > ### Author Response · Authors · 2024-09-23
> > > > > **Thank you!**
> > > > >
> > > > > Thank you so much for your timely response and for your efforts in providing such constructive feedback on our work!

---

### Decision · Action_Editor_ztnq · 2024-09-24

**Recommendation:** Accept as is

**Comment:**

The reviewers are all in agreement that the paper has substantial novelty and significance, and all remaining concerns/questions were addressed adequately during the discussion period.

**Audience:**

This paper is an excellent match to TMLR and is of broad interest to the ML community.

**Claims And Evidence:**

This paper introduces Reward Guided Latent Consistency Distillation (RG-LCD), which is a method for improving image generation quality in Latent Consistency Models by aligning outputs with human preferences, using a latent proxy reward model to address reward over-estimation.  The evidence is primarily experimental, showing cases such as a 2-step method of theirs beating a 50-step method of an existing approach.  Both human evaluation and automatic evaluation are considered, and ablation studies are also performed.